# Genomic Next-Token Predictors are In-Context Learners

Nathan Breslow      *nbreslo1@jh.edu*
Aayush Mishra      *amishr24@jh.edu*
Mahler Revsine      *mrevsin1@jh.edu*
Michael C. Schatz      *mschatz@jhu.edu*
Anqi Liu      *aliu.cs@jhu.edu*
Daniel Khashabi      *danielk@jhu.edu*
*Department of Computer Science, Johns Hopkins University*

Reviewed on OpenReview: *https://openreview.net/forum?id=KmNFx8DmaZ*

## Abstract

In-context learning (ICL) – the capacity of a model to infer and apply abstract patterns from examples provided within its input – has been extensively studied in large language models trained for next-token prediction on human text. In fact, prior work often attributes this emergent behavior to distinctive statistical properties in *human* language. This raises a fundamental question: can ICL arise *organically* in other sequence domains purely through large-scale predictive training?

To explore this, we turn to genomic sequences, an alternative symbolic domain rich in statistical structure. Specifically, we study the Evo2 genomic model, trained predominantly on next-nucleotide (A/T/C/G) prediction, at a scale comparable to mid-sized LLMs. We develop a controlled experimental framework comprising symbolic reasoning tasks instantiated in both linguistic and genomic forms, enabling direct comparison of ICL across genomic and linguistic models. Our results show that genomic models, like their linguistic counterparts, exhibit log-linear gains in pattern induction as the number of in-context demonstrations increases. To the best of our knowledge, this is the first evidence of organically emergent ICL in genomic sequences, supporting the hypothesis that ICL arises as a consequence of large-scale predictive modeling over rich data. These findings extend emergent meta-learning beyond language, pointing toward a unified, modality-agnostic view of in-context learning.

## 1 Introduction

Scaling Large Language Models (LLMs) has revealed an unexpected and powerful capacity: in-context learning (ICL) (Radford et al., 2018; Brown et al., 2020a), the ability to infer and apply abstract patterns purely from examples contained within their input. Unlike conventional adaptation, which relies on gradient updates, LLMs internalize mechanisms for flexible pattern induction (Olsson et al., 2022b), allowing them to perform few-shot generalization and analogical reasoning (Srivastava et al., 2023), *without explicit parameter updates.* This behavior, which arises organically from large-scale "next-token" prediction on human language, has been interpreted as *emergent* forms of meta-learning (learning to learn) within a *fixed* parametric system.

Almost all prominent evidence of emergent ICL so far (Srivastava et al., 2023, inter alia) comes from training on human language (e.g., English). This raises a fundamental question: *is there something inherently special about human language that enables ICL to emerge?* One can take two different stances on this:

- $H_1$: Human language possesses distinctive distributional properties such as parallelism or compositionality (Chen et al., 2024; Hahn & Goyal, 2023) that uniquely nurture ICL.

---

LLMs were used during the conception, implementation, and refinement of this paper. We assume full responsibility for all content.

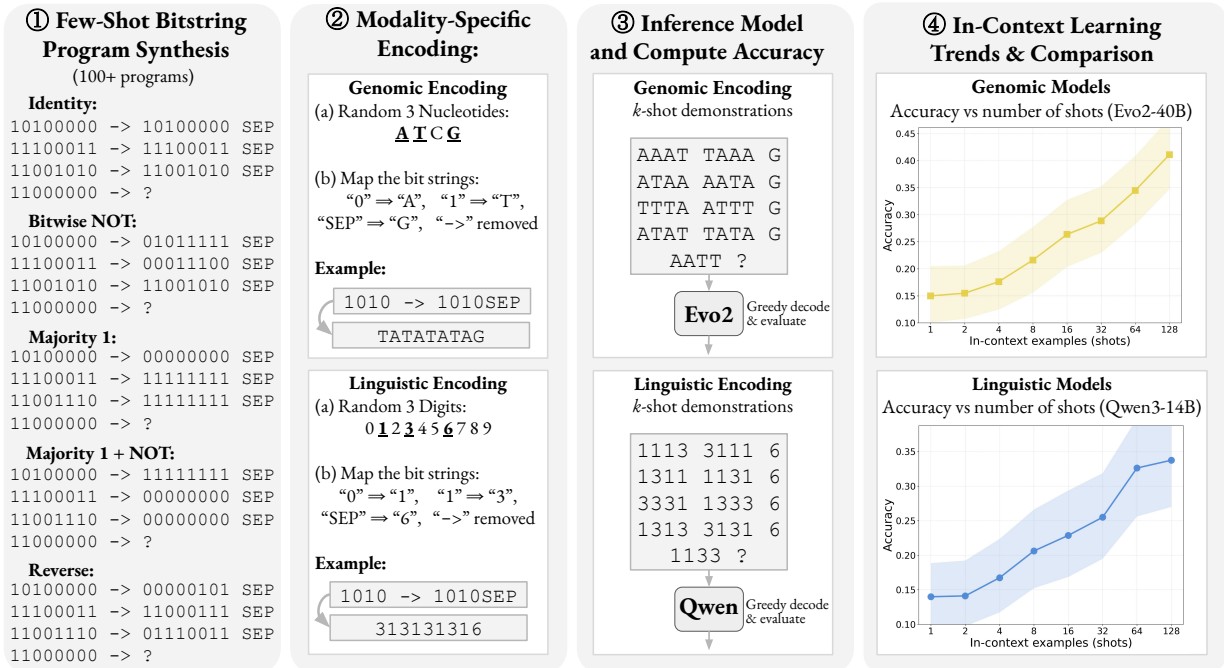

Figure 1: We design parallel symbolic reasoning tasks that allow direct comparison of ICL behavior across modalities (linguistic and genomic). ① Few-shot bitstring program-synthesis tasks (e.g., identity, NOT, majority, reverse) require models to infer mappings from examples. ② Each task is rendered in two modality-specific encodings: genomic (bitstrings mapped to random nucleotides A/T/C/G) and linguistic (bitstrings mapped to random digits), preserving abstract structure but differing in surface form. ③ Both genomic (Evo2) and linguistic (Qwen3) models receive $k$-shot demonstrations and are greedily decoded to compute exact-match accuracy. ④ **Both models show log-linear accuracy gains with more demonstrations**.

- **$H_2$:** Alternatively, perhaps natural language is merely a convenient substrate – having ample data and being naturally interpretable – and ICL could just as well arise in any sufficiently structured data domain, provided the model is large enough and trained to compress predictive patterns.

While many works have examined $H_1$, the veracity of $H_2$ remains far less understood. To this end, we turn to a radically different substrate: **the genome**. Genomic sequences can be viewed as a form of *natural* language that has evolved through nature. Like human language, they comprise complex symbol sequences that exhibit rich statistical regularities – motifs, repeats, dependencies (Benegas et al., 2025) that could, in principle, support pattern induction within context.

Testing for the existence of ICL requires large-scale models. Historically, genomic models were small and specialized, optimized for narrow biological tasks such as motif discovery or structure prediction. The advent of deep biological models (Ji et al., 2021) has expanded their scale, paralleling those of early LLMs. Most recently, the **Evo2** series (Brixi et al., 2025) marks a major step forward: a large-scale model trained solely on 'next-nucleotide' prediction. With training scale comparable to mid-sized LLMs (e.g., Qwen3-14B (Yang et al., 2025)), these genomic models finally make it possible to systematically compare genomic and linguistic representations under large-scale autoregressive training. If ICL arises primarily from scale and predictive compression ($H_2$), then large-scale genomic models such as Evo2 must already exhibit it *under our noses*. Like LLMs, they are trained for next-token prediction without any explicit meta-learning signal. Yet, despite extensive analysis of Evo2's biological capabilities (e.g., protein fitness prediction, variant effect estimation), its potential for emergent few-shot or analogical reasoning within context remains unexplored.

The second challenge in evaluating $H_2$ lies in developing a framework to detect and quantify ICL in the genomics domain where "tasks" lack natural human interpretability. Moreover, the task design must permit systematic comparison between ICL behaviors in human language and genomic modalities. To investigate

this, in §3 we introduce a controlled experimental framework (Fig. 1) for evaluating ICL across linguistic and genomic modalities. As illustrated in ①–②, we design a suite of symbolic bitstring program-synthesis tasks and render them in two parallel encodings: one using *genomic* alphabets (nucleotides: A/T/C/G) and one using *linguistic* (characters). We then evaluate both model families – Evo2 for genomic sequences and Qwen3 for linguistic text – under matched few-shot prompting conditions (③). This parallel encoding enables direct, apples-to-apples comparison of ICL scaling trends across genomic and linguistic models.

Our experiment's results (§4) reveal striking parallels: **both linguistic and genomic models exhibit log-linear improvements** in pattern induction and extrapolation as the number of demonstrations grows. To our knowledge, this is the first evidence of organically emergent ICL in genomic sequences. These findings suggest $H_2$, that is, *ICL is not an artifact that is specific to human language, but a broader consequence of compression and predictive modeling in pattern-rich sequence spaces*. By demonstrating ICL in a non-linguistic biological domain, we extend the scope of emergent meta-learning beyond natural language, pointing toward a unified view of context-adaptive computation that spans different modalities of data.

We further perform analyses to compare the types of tasks at which genomic and linguistic models excel. Our results reveal distinct patterns: genomic models learn faster with more demonstrations, while linguistic models gain more from scale (§4.2); genomic models perform best on direct copying or local transformations, whereas linguistic models excel on tasks requiring single-bit dependencies or global summary statistics (§4.3, §4.4). We note the caveat that these findings are specific to our experimental conditions (e.g., our choice of reasoning tasks) and may not be a holistic statement about ICL in genomic and linguistic models. We further validate these findings by demonstrating that Evo2 also exhibits robust few-shot learning on a real-world genomic promoter classification task (Appendix E).

**Our Contributions.** (a) We present a controlled experimental setup for detecting and measuring ICL across linguistic and genomic modalities. Our framework defines a suite of symbolic bitstring reasoning tasks that can be rendered in both representations, enabling direct, apples-to-apples comparison of ICL behavior. (b) Using the Evo2 family of large-scale genomic models, we demonstrate for the first time that models trained solely on nucleotide sequences exhibit clear in-context learning, mirroring the scaling trends of Qwen3 language models. (c) Our findings challenge the idea that ICL depends on linguistic structure. Instead, they point to ICL as a modality-agnostic outcome of large-scale next-token prediction over pattern-rich data, reflecting general principles of compression and contextual inference. Together, these results extend the scope of emergent meta-learning beyond language, toward a unified understanding of context-adaptive computation in artificial and biological systems alike.

## 2 Related Work and Broader Context

**Why does ICL remain interesting?** ICL remains a striking and conceptually rich phenomenon: it emerges *without* task-specific training yet enables rapid, in-situ adaptation. It comes closest to long-standing ambitions in classical AI—such as case-based (Aamodt & Plaza, 1994; Lancaster & Kolodner, 1987; Ross, 1984; Ellman, 1989) and analogical reasoning (Hofstadter, 2001; Gentner & Hoyos, 2017; Holyoak et al., 2001) where agents solve new problems by reusing solutions from prior (analogous) examples.

Beyond its theoretical appeal, ICL has become a practical workhorse in recent advances, e.g., synthetic data generation using pretrained models (Wang et al., 2023c; Shao et al., 2023; Tunstall et al., 2023) or reasoning (Wei et al., 2022; Nye et al., 2021; Yao et al., 2023; Yasunaga et al., 2023). In short, ICL is both a window into emergent learning dynamics and a cornerstone for building adaptive, interpretable systems.

**Emergent-ICL ≠ Meta-ICL:** A parallel line of research studies models trained with an explicit *ICL objective* – learning to predict over collections of input–output pairs $(x, f(x))$. For example, regression models that are trained *explicitly* to infer an underlying rule (e.g., linear, polynomial, or sinusoidal) from few-shot examples (Garg et al., 2022; Li et al., 2023c; Raventós et al., 2023; Nejjar et al., 2024). This setup mirrors *meta-learning*, where a model is explicitly "trained to learn" (Finn et al., 2017; Bertinetto et al., 2019; Zintgraf et al., 2019). We refer to such systems as *Meta-ICL* (Panwar et al., 2023; Min et al., 2022; Wu et al., 2022; Kirsch et al., 2022; Zhang et al., 2023), in contrast to *emergent ICL* in large pretrained models, which arises organically from next-token prediction over natural data.

Meta-ICL results demonstrate that transformers *can be optimized* to perform meta-learning but fail to explain why *emergent* ICL appears in models never trained for it. Critically, Meta-ICL generalizes poorly outside its training domain (Naim et al., 2024). For example, a Meta-ICL model trained on linear and cosine functions fails on their composition (Yadlowsky et al., 2023)—and does not capture higher-order abilities like in-context reasoning (Wei et al., 2022; Kojima et al., 2022) or adaptive self-refinement (Madaan et al., 2023) – unless explicitly trained for them. In contrast, emergent ICL in LMs generalizes broadly and flexibly across diverse problem types (Srivastava et al., 2023). These behavioral and mechanistic differences support our working assumption that *Meta-ICL* and *emergent ICL* are fundamentally distinct phenomena (Shen et al., 2024; Mishra et al., 2025). Throughout, our focus in this work is on the latter – the organic emergence of in-context learning in large pretrained models.

**ICL $\neq$ Zero-shot learning:** Existing literature in the domain of timeseries studies the zero-shot inference capabilities of next-token predictors, i.e., given a context of a given problem their models can predict the forthcoming values (tokens) (Ansari et al., 2024; Das et al., 2024). This differs from the type of ICL we evaluate, which explicitly relies on *few-shot* learning of a transformation, not zero-shot adaptation. In fact, in Fig. 2, we report that Evo2's performance begins far below a naive mode baseline (simply guessing the most common output) at one shot, indicating very poor zero-shot capability. Evo2's ability only becomes meaningful *only when many shots are provided*. This makes the type of ICL we demonstrate qualitatively different from zero-shot ability, as it explicitly only appears in few-shot contexts.

**Are there any prior results on emergent ICL in genomic models?** Most genomic models to date are *encoders* (Ji et al., 2021; Dalla-torre et al., 2024), which preclude autoregressive generation and the study of ICL in the standard sense. Among the few autoregressive genomic models, most remain small, e.g., GENA-LM (300M) (Fishman et al., 2024) and scGPT (53M) (Cui et al., 2024). The recently-released Evo2 series (Brixi et al., 2025) includes models up to 40B parameters, comparable in scale to modern LLMs. This model family provides the first viable testbed for probing *emergent* ICL, which we study here. A related effort, HyenaDNA (Nguyen et al., 2023), trains on million-token genomic sequences and reports ICL-like behaviors; however, these involve *meta-ICL* mechanisms (e.g., soft prompting and instruction fine-tuning; see their Sec. 4.3), rather than organically emergent ICL. Similarly, Bio-xLSTM (Schmidinger et al., 2024) demonstrates ICL, but only after explicitly training on deliberately constructed few-shot demonstrations instead of natural biological data - another example of Meta-ICL. An additional avenue of research demonstrates that genomic models can perform a form of ICL by copying prior tokens in-context for prediction (Kantroo et al., 2025) – akin to how induction heads function (Olsson et al., 2022a). This, however, doesn't demonstrate any ability to infer latent functions or apply a novel rule in-context.

**Are there instances of emergent ICL in other non-linguistic modalities?** While transformers have been extended to many modalities beyond language, genuine cases of *emergent* ICL arising purely from large-scale pretraining remain rare. In neuro-signal modeling, for instance, EEG-GPT (Kim et al., 2024) is trained with explicit in-context demonstrations, making it an instance of *Meta-ICL* rather than emergent adaptation. Similarly, multimodal models such as Flamingo (Alayrac et al., 2022), BLIP/BLIP-2 (Li et al., 2022; 2023a), and Emu (Sun et al., 2023; 2024) exhibit in-context behavior only because they are *explicitly trained* to process demonstration patterns – again, meta-learned rather than emergent. Another example of non-linguistic ICL is Chronos 2 – a timeseries model trained to perform in-context learning through cross-batch learning, where the model's performance can be improved by learning from multiple time series provided in-context. However, Chronos 2 is trained using exclusively synthetic data designed to elicit cross batch learning and ICL, which firmly places it in the Meta-ICL regime (Ansari et al., 2025). To our knowledge, the only claimed instance of emergent ICL outside language is in vision: Bai et al. (2024) show that transformers pretrained on natural visual sequences can infer analogical and compositional tasks without explicit supervision. The weakness of this result is that their ICL prompts consist of long sequences of contiguous frames (Bai et al., 2024, Fig 10), which effectively reduces the task to next-frame prediction, rather than true ICL that requires inferring compositions of multiple demonstrations.

# 3 Experimental Framework for Cross-domain In-context Learning

This section presents our framework for evaluating ICL across linguistic and genomic models. We outline the experimental desiderata (§3.1), setup (§3.2), model selection (§3.3), and evaluation protocol (§3.4). We additionally release code for replicating these experiments on Github.

## 3.1 Experimental Desiderata

**Desideratum 1: Cross-domain comparability.** Our experimental framework requires that each task be performable by *both* language and genomic models. Thus, every task must be representable in both linguistic and nucleotide alphabets. This constraint excludes existing benchmarks that rely on domain-specific semantics – such as language reasoning datasets (e.g., BIG-Bench, StrategyQA, GSM8K (Srivastava et al., 2023; Geva et al., 2021; Cobbe et al., 2021)) or biological tasks (e.g. variant effect prediction, exon identification (Brixi et al., 2025)). While one could, in principle, translate domain-specific tasks into an alternate alphabet (e.g., mapping language tokens to base sequences via quaternary encoding), doing so inherently biases the evaluation: the source domain retains advantages, while the target domain must operate on representations that are unnatural to it. As a result, such cross-domain encodings confound the comparison, reflecting representational translation artifacts rather than genuine differences in ICL behavior.

**Desideratum 2: Limited vocabulary.** Since genomic models operate on only four nucleotides (A, T, C, G), tasks must be expressible within an equally compact alphabet. This rules out the existing symbolic reasoning and analogy benchmarks (Hodel & West, 2023; Lewis & Mitchell, 2025; Webb et al., 2025), which rely on richer vocabularies or background knowledge (e.g., shapes, colors, or linguistic tokens with explicit semantic/lexical roles). While some of these tasks are more symbolic in nature, they still rely on pretrained knowledge or linguistic instructions about numerical or lexical ordering (i.e. 'e' coming after 'a b c d') – information that couldn't effectively be conveyed in-context to a genomic model (Lewis & Mitchell, 2025; Stevenson et al., 2025). Such tasks cannot be faithfully represented in a four-token regime without introducing artifacts or structural loss. Accordingly, our evaluation focuses on tasks that retain abstract reasoning structure while remaining compatible with the low-vocabulary symbolic space shared across linguistic and genomic models.

## 3.2 Experimental Setup

**Notation:** We formally define ICL as follows. Let $S$ be the input domain and $O$ the output domain, and let a task be a latent deterministic function $f : S \to O$. A pretrained autoregressive model $M$ performs ICL by conditioning on an ordered $n$-shot demonstration set $E = \big((x_1, f(x_1)), \ldots, (x_n, f(x_n))\big)$, where $x_i \in S$, and then receiving a held-out query $x \in S \setminus \{x_1, \ldots, x_n\}$. Conditioned on $(E, x)$, the model produces a prediction $\hat{y} = M(E, x)$, which we compare to the ground-truth $f(x)$. Then the single-trial success is $\mathbb{1}\big[\hat{y} = f(x)\big]$.

**Task Formulation: in-context program induction over abstract symbols.** Given our desiderata (§3.1), we design symbolic induction tasks that require a small lexicon. Each task requires a model to infer a hidden transformation rule from a few input–output examples and apply it to a new query. We refer to this process as *program synthesis in-context*. This setup isolates in-context learning ability by removing the influence of pretrained world knowledge. If the function family $f$ and symbol space $S$ are chosen carefully, the evaluation is far from the model's pretraining distribution. Similar formulations have been used in prior work on compositional reasoning (Brown et al., 2020b), analogical reasoning via Raven's Progressive Matrices (Raven et al., 1962; Webb et al., 2023), and more recently in ARC-AGI (Chollet, 2019).

**Bitstring domain and function space.** To maintain compatibility across genomic and linguistic models, we represent all symbols as bitstrings of length $k$, i.e., $S = \{0, 1\}^k$. Genomic models operate over four nucleotides (A, C, G, T), allowing two tokens (e.g., A/C) to encode 0/1 and reserving the others (G/T) as delimiters or separators. Bitstrings are also naturally supported by linguistic models since most tokenizers represent digits as single tokens, ensuring parity in symbol granularity. This design minimizes tokenization confounds while providing a uniform symbolic substrate for comparing both domains. See Fig. 1 for examples. Despite their simplicity, bitstrings support a wide range of composable operations – from basic (identity,

constant) to logical (AND, OR), positional (shift, rotation, reversal), and aggregate (majority, parity). This expressivity makes them a convenient and extensible testbed for cross-domain in-context learning evaluation.

We set the symbol space to $S = \{0, 1\}^8$, corresponding to all 8-bit binary strings. This choice balances expressivity (256 unique bitstrings) with manageable prompt length. To define the task space, we construct a set of 100 transformation functions $F \subseteq \{f \mid f : S \to S\}$, where each function maps one bitstring to another according to a deterministic rule. Each $f \in F$ is either a single primitive operation or a composition of two primitives, allowing for a controlled yet diverse range of symbolic transformations.

**Primitive operations.** We build $F$ from a library of 30 fundamental primitives (listed in Appendix A.1) spanning six functional categories: (1) *Bitwise transformations* (e.g., flipping individual bits), (2) *Structural rearrangements* (rotations, shifts, reversals, swaps), (3) *Positional masking and selection* (preserving or zeroing specific bit positions), (4) *Pattern detection* (detecting alternating or palindromic patterns), (5) *Aggregation functions* (majority/minority, parity), and (6) *Trivial mappings* (identity, constant outputs). This taxonomy ensures coverage of both low-level logical operations and higher-order compositional structure.

**Automatic generation and validation.** To construct the final function set, we use GPT-5-Codex to automatically generate 100 unique transformations by composing primitives into valid Python programs. Each candidate function is verified programmatically for *logical distinctness*: no two programs $f, g \in F$ produce identical outputs for all inputs ($\nexists f, g$ such that $f(x) = g(x)$ for all $x \in S$). All functions undergo manual inspection to confirm correctness, diversity, and adherence to the intended categories. The full list of transformations and corresponding source code is provided in Appendix A.2.

**Prompt encoding.** Prompts are encoded using a unified symbolic scheme to enable evaluation across linguistic and genomic models. For linguistic models, bits $(0, 1)$ are mapped to two random digits (0–9); for genomic models, to random nucleotides $(A, T, C, G)$. The separator token $(\to)$ is omitted, and in-context examples are separated by a randomly chosen unused token distinct from those representing 0 and 1. (See Fig. 1 for examples.) All mappings are randomized per trial to avoid memorization or positional bias.

### 3.3 Model Families and Selection Rationale

For fair cross-domain comparison, we use two representative model families: Qwen3 for human language and Evo2 for genomics. The rationale for this selection is as follows:

(a) *Parameter scaling:* Both families span multiple orders of magnitude in parameter count, enabling systematic scaling analysis of ICL ability. The Qwen3 series ranges from 0.6B to 14B parameters, while Evo2 includes 1B, 7B, and 40B models (Yang et al., 2025; Brixi et al., 2025). This parallel scaling structure facilitates consistent measurement of how ICL performance evolves with model size across linguistic and genomic modalities.

(b) *Compute matching:* The largest models in each family are trained with comparable total compute, offering an opportunity for an approximately compute-matched cross-domain comparison. Using the standard $6ND$ estimate (Kaplan et al., 2020), Qwen3-14B-Base is trained with about $3.2 \times 10^{24}$ FLOPs, while Evo2-40B is trained with $2.25 \times 10^{24}$ FLOPs (Yang et al., 2025; Brixi et al., 2025), making Evo2 uniquely well-suited for comparison with Qwen3 at scale.

(c) *Availability of base models:* The Qwen3 family releases base (*pre*-instruction-tuned) models at all scales up to 14B parameters, enabling direct evaluation of the intrinsic inductive reasoning ability of pure next-token predictors, without instruction-tuning artifacts. The Evo2 base models have no additional instruction tuning applied—the only nuance is that Evo2 1B is trained at a context length of 8192 nucleotides, whereas the 7B/40B are extended to a context length of one million.

(d) *Tokenizer:* Qwen3 uses a standard BPE tokenizer with a vocabulary size of 151,669. Evo2 uses a byte-level tokenizer, so individual nucleotides are mapped to individual tokens. Notably, Qwen's tokenizer maps single digits to single tokens, allowing for parity in our experiments (Yang et al., 2025; Brixi et al., 2025).

(e) *Context length:* Qwen3's 0.6B and 1.7B models have a context length of 32K. The remaining dense models have a context length of 128K. Evo2's 1B model has a context length of 8K, and the remaining models have a context length of 1M. As none of our experiments approach these limits, we can use all models in both families without fear of context length as a confound (Yang et al., 2025; Brixi et al., 2025).

(f) *Training corpora:* All Qwen3 models are trained on 36 trillion text tokens covering a wide variety of topics and languages. Evo1 1B is trained on 1 trillion tokens, Evo2 7B is trained on 2.4 trillion tokens, and Evo2 40B is trained on 9.3 trillion tokens. These extensive training corpora ensure that any potential ICL dynamics have thoroughly emerged (Yang et al., 2025; Brixi et al., 2025).

(g) *Architecture:* Qwen3 is based on a conventional Llama-like transformer architecture, while Evo2 uses the StripedHyena2 architecture which intersperses convolutional layers with attention-based ones. While ideally both model families would use the same architecture, there were no vanilla transformers at Evo2's compute scale (Yang et al., 2025; Brixi et al., 2025).

(h) *Licensing:* All models are released under Apache 2.0 (Yang et al., 2025; Arc Institute, 2025a;c;b), ensuring reproducibility.

### 3.4 Evaluation Protocol

**Evaluation metric.** Building on the setup in §3.2, we now describe how in-context performance is evaluated. For each transformation $f \in F$, we draw an $n$-shot demonstration set $E \subset S$ and a held-out query $x \in S \setminus E$. The model $M$ receives a few-shot prompt of input–output pairs followed by the query and produces a prediction $\hat{y} = M(E, x)$. A trial is marked correct if $\hat{y} = f(x)$, giving the single-trial indicator $\mathcal{A}(f, E, x, M) = \mathbb{1}\big[\hat{y} = f(x)\big]$.

Since exact enumeration of *all* the programs is intractable, we use a Monte Carlo estimate (a full formulation of what we approximate can be found in the appendix B). Let $E_n$ denote the space of all sets of input bitstrings with cardinality $n$: $E_n = \{E \subset S : |E| = n\}$. Sample $m$ i.i.d. context sets $E^{(t)} \sim \mathrm{Unif}(E_n)$ for $t = 1, \ldots, m$; for each $t$, sample $x^{(t)} \sim \mathrm{Unif}(S \setminus E^{(t)})$. Then our empirical estimate of accuracy, for a given transformation $f$ is $\hat{\mathcal{A}}_f(M, n) = \frac{1}{m} \sum_{t=1}^{m} \mathcal{A}\big(f, E^{(t)}, x^{(t)}, M\big)$ and the overall estimated accuracy for model $M$ is:

$$\hat{P}(M, n) = \frac{1}{|F|} \sum_{f \in F} \hat{\mathcal{A}}_f(M, n). \tag{1}$$

**Model suites and sampling configuration.** We fix the number of Monte Carlo trials to $m = 8$. The set of evaluated *genomic models* is $\mathcal{M}_G = \{\texttt{evo2\_1b\_base}, \texttt{evo2\_7b}, \texttt{evo2\_40b}\}$ and the set of *linguistic models* is $\mathcal{M}_L = \{\texttt{Qwen3-0.6B-Base}, \texttt{Qwen3-1.7B-Base}, \texttt{Qwen3-4B-Base}, \texttt{Qwen3-8B-Base}, \texttt{Qwen3-14B-Base}\}$. We evaluate across shot counts $\mathcal{N} = \{1, 2, 4, 8, 16, 32, 64, 128\}$. For each model $M$ and shot count $n$, we compute:

$$P_L = \{\hat{P}(M, n) : M \in \mathcal{M}_L, n \in \mathcal{N}\}, \quad P_G = \{\hat{P}(M, n) : M \in \mathcal{M}_G, n \in \mathcal{N}\}.$$

We estimate standard errors via a two-stage nonparametric cluster bootstrap, resampling functions (clusters) and within each selected function, resampling its evaluation samples with 5000 replicates for each $(M, n)$.

**Mode baseline.** We define a mode baseline that always predicts the most frequent output observed in the context. For a given function $f$, context set $E \subset S$, and query input $x \in S \setminus E$, the mode prediction is $\hat{y}_{\mathrm{mode}}(f, E, x) = \arg\max_{y \in S} \big|\{e \in E : f(e) = y\}\big|$, where ties in the $\arg\max$ are broken randomly. This simply corresponds to guessing the most common output in the few-shot examples. The overall mode baseline with $n$ shots and across the set of all functions $F$ is:

$$\hat{P}_{\mathrm{mode}}(n) = \frac{1}{m|F|} \sum_{f \in F} \sum_{t=1}^{m} \mathbb{1}\big[\hat{y}_{\mathrm{mode}}(f, E^{(t)}, x^{(t)}) = f(x^{(t)})\big], \tag{2}$$

where $E^{(t)}$ and $x^{(t)}$ are sampled identically to the model evaluation in Eq.1. This baseline corresponds to making an educated guess based only on the overall distribution of function outputs that simply learns the majority statistics of the prompt without attempting to infer the underlying transformation.

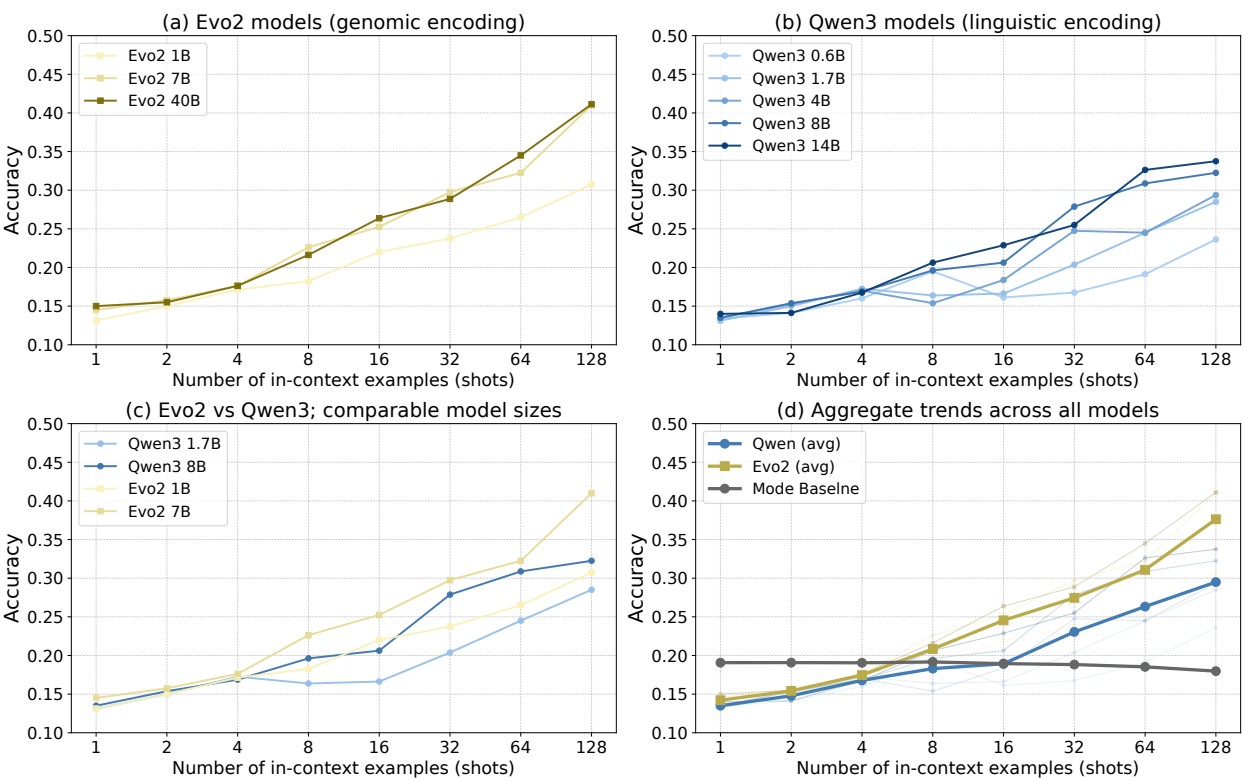

Figure 2: Few-shot performance of Qwen3 and Evo2 models. (a) Evo2 model performance with respect to log(shots). All models monotonically improve – the 7B and 40B have roughly equivalent performance, and the 1B trails behind them. (b) Qwen3 model performance with respect to log(shots). All models improve, but not always monotonically. Smaller models struggle in 4-16 shot range. (c) At comparable sizes, Evo2 outperforms Qwen3. (d) Averaged performance across both model families shows consistent improvement with respect to log(shots). All models exceed the mode baseline shown in gray color. The exact accuracies and error bars (bootstrap-based standard errors) are included in Appendix C.

## 4   Empirical Results

This section reports empirical findings on few-shot bitstring generalization. §4.1 presents the main accuracy trends with respect to model size and shot count. §4.3 examines sensitivity to task complexity using a BitLoad measure, and §4.4 contrasts the models' qualitative competencies across individual transformations.

### 4.1   Main Results

**Across both model families, accuracy generally rises linearly with respect to** log(**shots**)**.** A linear regression linking performance to log(shots) yields highly significant slopes for all models (all $p \leq 10^{-3}$ via one-sided t-test on slope). As Fig. 2 shows, Evo2 models show cleanly monotonic gains, with a pronounced step from 1B to 7B, and near-indistinguishable curves for 7B and 40B; by 128 shots, both 7B and 40B surpass 40% accuracy (Fig. 2a). Qwen3 also trends upward overall but with non-monotonic patches—especially for smaller models in the 4–16 shot band—before resuming clear improvements from 32 to 128 shots; at 128 shots, the 14B model approaches 35% (Fig. 2b). A full table of all accuracies is in Appendix C.

**Evo2 outperforms Qwen3 at comparable sizes.** Evo2 1B beats Qwen3 1.7B and Evo2 7B beats Qwen3 8B at higher shot counts (Fig. 2c). Averaging within families reinforces the same picture: comparable performance in the 1–4 shot range, Evo2 pulling ahead around 8–16 shots, and the gap persisting at higher shot counts (Fig. 2d).

**The mode baseline's performance doesn't improve with more shots, and lags Qwen3 and Evo2 at high shot counts** (Fig. 2d). This indicates that the in-context learning performed by Qwen3 and Evo2 is qualitatively different than simply sampling from the distribution of possible outputs in-context. Qwen3 and Evo2's performance requires that they learn, in-context, how to condition on the input. Furthermore, for Qwen3, advantages over the mode baseline become consistently significant ($p < 0.05$ via one-sided z-test on boot-

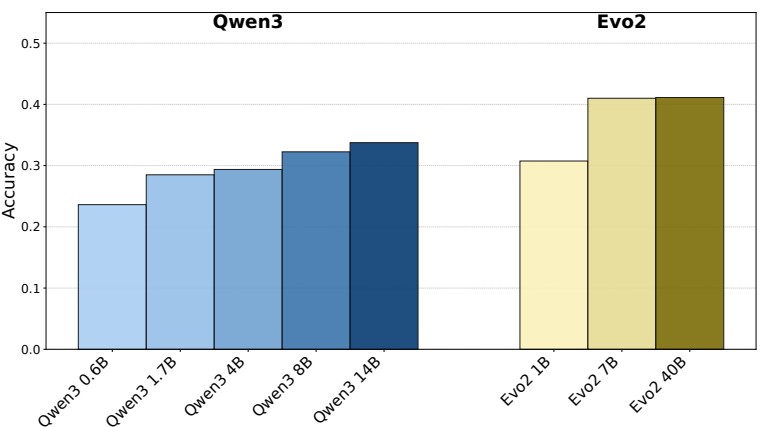

Figure 3: Performance at $n = 128$ shots. All model accuracies increase monotonically with respect to parameter count.

strapped standard errors) at $n=128$ for all sizes greater than 0.6B. For Evo2, statistically significant advantages emerge slightly earlier: for 1B at $n=64$, the 7B at $n = 32$, and the 40B at $n = 16$. Regardless, all models surpass the naive baseline.

## 4.2 Analysis of ICL Capability Across Model Families and Sizes

As shown in Fig. 3, **both Qwen3 and Evo2 exhibit clear performance gains with increasing scale**. Qwen3 displays a strong positive correlation between accuracy and parameter count ($p < 0.05$ for all shot counts $\geq 16$, one-sided $t$-test on slope), while Evo2 shows a distinct jump from 1B to 7B but little improvement beyond that. This suggests that in-context program induction becomes more robust with scale, though saturation may occur once models reach sufficient capacity (A detailed statistical analysis in Appendix D).

## 4.3 ICL Sensitivity to Task Complexity: BitLoad Analysis

To understand which transformations Qwen3 and Evo2 infer most effectively, we analyze their performance across varying task complexities. We focus on the largest models in each family (Qwen3-14B and Evo2-40B) under the ($n = 128$) shot regime, providing both models ample opportunity to display their ICL abilities.

**Defining BitLoad.** We introduce *BitLoad*, a measure of a function's intrinsic complexity. Informally, BitLoad quantifies how many input bits influence the output. Formally, it is defined as:

$$\text{BitLoad}(f) = \sum_{i=1}^{k} \mathbb{1}\Big[\exists x, j : f_j(x) \neq f_j(x^{\oplus i})\Big], \quad (3)$$

where $x^{\oplus i}$ denotes $x$ with bit $i$ flipped, and $f_j(x)$ returns the $j$-th output bit. Intuitively, it counts the number of bit positions whose perturbation changes the output. So, the larger the BitLoad of a function, the harder it is since it requires the model to attend to more bits. This metric is similar to the existing statistical measure of "relevant features", just defined specifically for our bitstring manipulation tasks. (Nevo & El-Yaniv, 2002).

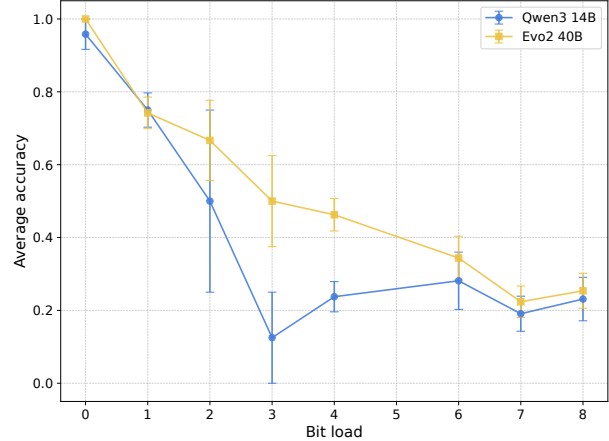

Figure 4: Accuracy vs. BitLoad averaged across all tasks (BitLoad; Eq.3). Qwen declines sharply, while Evo degrades more gradually. Details in §4.3.

Fig. 4 shows the mean accuracy of Qwen vs Evo with respect to the BitLoad of all of our tasks. A full table of the BitLoad of every tested task is attached in the appendix A.3. We see that both models achieve near-perfect accuracy on constant (0 BitLoad) tasks and remain similar at BitLoad 1. However, by BitLoad 2,

Evo begins to outperform Qwen, and beyond that point Qwen's accuracy drops sharply – falling below 20% by BitLoad 4 – while Evo's performance declines more gradually, remaining above 40% before converging near 20% at BitLoad 8. This asymmetry suggests that Evo maintains partial generalization as dependency depth increases (BitLoad value), whereas Qwen's ICL collapses more abruptly.

While BitLoad strongly correlates with task accuracy, Fig. 4 also shows notable deviations within the error bars. Thus, while task complexity is a strong predictor of ICL accuracy, other factors (e.g., transformation depth, pretraining exposure) likely play meaningful roles, which motivate our qualitative study in §4.4.

### 4.4 Qualitative Analysis of Functional and Behavioral Differences

To complement the quantitative BitLoad analysis (§4.3), we conduct a per-task comparison to identify qualitative differences between Qwen3 and Evo2. Specifically, we analyze which tasks each model performs best on and how their inductive profiles diverge. We focus on the $n = 128$ shot regime for the largest models – Qwen3-14B and Evo2-40B – where both have maximal opportunity to exhibit ICL. We rank all tasks by model accuracy and examine the top ten for each model. Tasks that appear in one model's top ten but not the other are considered "exclusive."

**Exclusive competencies.** Qwen's exclusive tasks involve right-shift operations: `"spread_last_bit"` → `"shift_right_zero"` and `"edge_mask"` → `"shift_right_zero"`. For instance, `"shift_right_zero"` pads the bitstring on the left with a zero and truncates the last bit (e.g., 01010000 → 00101000). Qwen achieves 100% accuracy on `"spread_last_bit"` → `"shift_right_zero"`, whereas Evo2 achieves only 50%. A similar but smaller gap appears for `"edge_mask"` → `"shift_right_zero"` (87.5% vs. 62.5%).

In contrast, Evo2's exclusive strengths involve multi-bit transformations. A clear example is `"flip_bits"` → `"right_half"`, which applies bitwise NOT followed by masking the first half of the input (e.g., 01011100 → 00000011). Evo2 achieves 87.5% accuracy, while Qwen only 25%. This task has a BitLoad of four, consistent with Evo's superior performance on medium-complexity (2–4 bit) transformations observed in Fig. 4.

**Shared strengths.** Despite these differences, 7 of the top 10 tasks are shared between Qwen and Evo, yielding an intersection-over-union of 0.54. Both models excel at simple transformations such as constant-outputs or single-bit dependencies, e.g., `"spread_last_bit"` which copies the final bit to all positions.

**Differential skill profiles.** To sharpen contrasts, we identify the ten tasks most favoring each model. Qwen's advantages are concentrated in simple shifts and aggregation tasks. It outperforms Evo by 37.5% on the `"minority"` operation (output all 1s if zeros > ones), and by a similar margin on two parity-based tasks requiring counting the number of 1s. These trends suggest that Qwen may be better at reasoning over global properties of bitstrings. Its superiority on simple shifts may also be explained by the extreme rarity of frame shift mutations in DNA due to how catastrophic they are – a single nucleotide offset can decimate an entire protein. This is empirically supported by the fact that single-nucleotide deletions/shifts increase perplexity far more than other common mutations when presented to Evo2 (Brixi et al., 2025).

Evo2, by contrast, dominates tasks requiring full-bitstring manipulation. It achieves 62.5% on bitwise NOT (vs. 0% for Qwen), 62.5% on identity (vs. 12.5%), and large margins on compositions such as `"flip_bits"` → `"right_half"` (87.5% vs. 25%) and `"rotl1"` → `"flip_bits"` (37.5% vs. 0%). This exposes perhaps the most important difference between Qwen and Evo's ICL in this specific context: Evo can learn simple full-bitstring operations in-context, whereas Qwen cannot. Notably, Qwen3's base models are trivially capable of learning the identity in a more familiar few-shot context – when examples are presented with arrows and newlines separating them, instead of our intentionally unfamiliar encoding. Thus these results should be taken as an existence proof of ICL in Evo2, not a definitive statement of Evo2 having more ICL ability than Qwen. We leave a comparison of these models across broader tasks to future work.

## 5 Discussion

**What are the implications of our findings on prior efforts to explain the emergence of ICL?** First, let us organize the existing frameworks for pinpointing the conditions under which ICL emerges:

($E_1$) **ICL's emergence is due to data distributional properties:** The distributional properties of data, such as "parallel structures" in human language pretraining data (Chen et al., 2024), its compositional structure (Hahn & Goyal, 2023), "burstiness" (Chan et al., 2022) and other such properties (Wibisono & Wang, 2024; Reddy, 2023) may be of importance (and perhaps necessary) for the emergence of ICL.

($E_2$) **ICL's emergence is due to a compression mechanism:** The large-scale compression mechanism during massive pretraining might drive ICL (Elmoznino et al., 2024a;b; Hahn & Goyal, 2023).

($E_3$) **ICL's emergence may require specific architectural properties:** While Transformers might be better suited for ICL than LSTMs (Xie et al., 2021), evidence is mixed (Lee et al., 2023), and non-Transformer models have also demonstrated ICL capabilities (Grazzi et al., 2024; Park et al., 2024).

Our findings refine existing hypotheses about ICL's origins. The emergence of ICL in genomic models challenges accounts that rely solely on language-specific distributional structures ($E_1$). The presence of ICL across both genomic and linguistic models supports the compression-based explanation ($E_2$), suggesting that large-scale sequence compression and its induced inductive biases drive ICL across modalities. With respect to architecture ($E_3$), Evo2 – an autoregressive hybrid combining convolutional and attention layers rather than a pure Transformer – exhibits similar scaling behavior, indicating that ICL does not depend on the pure Transformer form. Instead, architecture provides an expressive substrate that enables pattern induction once exposed to sufficiently large and structured data. Overall, these results position ICL as a modality-agnostic outcome of large-scale next-token prediction, rather than a phenomenon tied to linguistic statistics or a specific architecture.

**What are the implications of our findings on the frameworks to explain how ICL operates?** We next consider the major perspectives that seek to explain how ICL operates. One view holds that ICL functions as a mix of *task learning* and *task retrieval*, with demonstrations serving either to recall pretrained capabilities or to enable learning on the fly (Pan et al., 2023; Lin & Lee, 2024; Wang et al., 2024; Fang et al., 2025). Our symbolic reasoning tasks, instantiated in both linguistic and genomic domains, provide direct evidence for this *task learning* mode, aligning with this hypothesis and prior work (Pan et al., 2023; Fang et al., 2025). Because these tasks do not depend on pretrained semantic priors, they do *not* invoke task *retrieval*, offering limited insight into the Bayesian view that interprets ICL as implicit inference over latent concepts (Xie et al., 2021; Panwar et al., 2023; Wang et al., 2023b; Jiang et al., 2024). Meanwhile, our results remain agnostic toward the optimization-based hypothesis, which posits that ICL implements an implicit gradient-descent-like process (Akyürek et al., 2022; Ahn et al., 2023; Mahankali et al., 2023; Li et al., 2023b), as well as the induction-based account, which attributes ICL to specialized "circuits" for performing inductive generalization (Elhage et al., 2021; Olsson et al., 2022b; Wang et al., 2023a; Bansal et al., 2023; Ren et al., 2024). Together, our results most strongly support the presence of genuine task *learning* within ICL.

**Does Evo2 have an innate advantage on these tasks?** Possibly, for multiple reasons. First, though Evo2's is trained on less tokens total, all of Evo2's training tokens are long sequences of repeated nucleotides, and very little of Qwen3's training tokens are long sequences of repeated digits. Second, Evo2's Striped-Hyena2 architecture was found to to significantly outperform a vanilla transformer on long DNA sequences (Brixi et al., 2025). This could give Evo2 an innate advantage on long contexts containing the same few symbols vs Qwen3. Our result results do not imply Evo2's ICL ability is superior to Qwen3's, and we concede that our few-shot prompting setup may be biased toward Evo2. Attempts to make the task more legible to Qwen3, however, ran into the confound that Qwen3 has pretraining exposure to bitstring manipulation. Any prompting setup that included 0s and 1s immediately resulted in an extreme increase in Qwen3's performance. Future work is needed to establish a method that controls for Evo2's structural advantages without granting Qwen3 an unfair edge via its pretraining knowledge.

**Why not test the models on semantic tasks?** Semantic tasks – such as identifying the capitals associated with countries, classifying malformed proteins, etc. – require a fair amount of pretraining exposure to the concepts involved in the task as well as measuring ICL. While it's undoubtedly ICL when a model infers a semantic transformation (for instance, country→capital, word→opposite), the pretraining knowledge necessary to manifest this ICL precludes its use in extreme cross-modality comparisons. Focusing on far simpler bitstring transformations that can be learned entirely in-context allows for an apples-to-apples comparison between Evo2 and Qwen3. To ensure that the ICL observed is not an artifact of this simplified domain, we

additionally demonstrate in Appendix E that Evo2 exhibits robust, scaling ICL on a native genomic task (promoter classification), though we exclude this from the main comparison to maintain parity with the linguistic models.

**Beyond the $H_1$-$H_2$ dichotomy.** While we presented $H_1$ and $H_2$ (in §1) as contrasting hypotheses, they are not necessarily mutually exclusive. It is entirely plausible that both hold simultaneously – that certain distributional properties inherent in natural data, shared across human language and other natural domains such as genomics, contribute to the emergence of ICL. In this view, linguistic compositionality may represent one instance of a broader statistical substrate that fosters ICL. We framed the two hypotheses as a dichotomy primarily to highlight the contrast between language-specific and modality-general explanations, rather than to suggest that only one can be true.

## 6 Conclusion

We introduce a suite of bitstring reasoning tasks that can be encoded in both natural language and genomic sequences, showing that genomic models – like their linguistic counterparts – exhibit clear in-context learning. Across all Evo2 model sizes, we observe robust log-linear gains in accuracy with increasing demonstrations, paralleling the scaling trends of Qwen3 language models. These findings challenge the notion that ICL is unique to human language, suggesting it emerges whenever an expressive model is trained autoregressively on structured, pattern-rich data.

**Potential future work:** This proof-of-existence for ICL lays the groundwork for many future research directions. One could broaden the test suite to incorporate tasks beyond bitstrings – up to four symbols are natively available in nucleotides, and far more can be used if one encodes at the codon level.

Another promising direction is mechanistic interpretability: analyzing Evo2's internal activations to identify which circuits enable ICL and how the model aggregates context to predict the next nucleotide. Comparing these mechanisms to known induction circuits (Elhage et al., 2021) in LLMs could reveal whether analogous structures exist across architectures – a question that connects biological and linguistic ICL dynamics.

Finally, this work motivates searching for ICL in other non-linguistic modalities – time series (Das et al., 2024), system logs (Akhauri et al., 2025), physics simulations (Holzschuh et al., 2025), chess games (Ruoss et al., 2024), and climate projections (Duncan et al., 2025). Each offers a structured, patterned substrate that could support its own form of contextual reasoning. These diverse modalities, each with their unique structure and constraints, suggest a rich world of non-linguistic ICL capability waiting to be explored, and this work represents a maiden voyage into these extremely interesting waters.

### Broader Impact Statement

Understanding ICL is pivotal for both the scientific study and practical control of LLMs. ICL ties to major efforts to interpret model behavior, characterize how reasoning and abstraction evolve with scale, and design mechanisms for controllability and steering. It also powers pragmatic applications such as synthetic data generation. By demonstrating that ICL arises even in non-linguistic settings, this work broadens the empirical foundation for studying ICL as a general computational phenomenon rather than a quirk of human-language training. A clearer mechanistic understanding of how ICL emerges and operates across modalities can inform how we build, guide, and evaluate large models – improving their reliability, controllability, and usefulness while reducing risks from misalignment or overgeneralization.

### Acknowledgment

This work is supported by ONR grant (N0001424-1-2089). We acknowledge the use of computational resources on the DSAI cluster. We sincerely thank the JHU CLSP and DSAI communities for their helpful comments and feedback.

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

# A    Further Details on Synthetic Task Definition

## A.1    Table of Primitives

The following table describes the thirty primitives used to construct the task space via composition – their use is described in §3.2.

| Primitive | Description |
|---|---|
| alternating_start_one | Produce a mask marking positions that differ from the alternating 1010... pattern starting with 1 (1 = mismatch, 0 = match). |
| alternating_start_zero | Produce a mask marking positions that differ from the alternating 0101... pattern starting with 0 (1 = mismatch, 0 = match). |
| center_mask | Zero out the first and last bits while leaving the interior bits unchanged; strings of length $\leq 2$ become all zeros. |
| double_rotl | Circularly rotate the bitstring two positions to the left. |
| double_rotr | Circularly rotate the bitstring two positions to the right. |
| edge_mask | Preserve the first and last bits and zero out every interior bit (length 0/1 strings pass through). |
| flip_bits | Invert every bit, swapping 0s for 1s and vice versa. |
| identity | Return the bitstring unchanged. |
| invert_prefix | Flip the bits in the left half of the string, keep the right half as is. |
| invert_suffix | Keep the left half as is and flip every bit in the right half of the string. |
| keep_even_positions | Keep bits at even indices (0-based) and zero out bits at odd indices. |
| keep_odd_positions | Keep bits at odd indices (0-based) and zero out bits at even indices. |
| left_half | Preserve the left half of the string and replace the right half with zeros. |
| majority | Fill the string with the majority bit from the input; ties resolve to all 1s. |
| meta_constant | Returns a random, pre-set constant. |
| minority | Fill the string with the minority bit from the input; ties resolve to all 0s. |
| mirror_half | Copy the left half of the string onto the right half in reverse order, keeping the center bit unchanged for odd lengths. |
| ones_if_palindrome | Output all 1s if the input is a palindrome; otherwise output all 0s. |
| parity_fill | Output all 1s when the input contains an odd number of 1s; otherwise output all 0s. |
| reverse_bits | Reverse the order of the bits in the string. |
| right_half | Zero out the left half and keep the right half unchanged. |
| rotl1 | Circularly rotate the bitstring one position to the left. |
| rotr1 | Circularly rotate the bitstring one position to the right. |
| shift_left_zero | Shift the string left by one, dropping the first bit and appending a 0 on the right. |
| shift_right_zero | Shift the string right by one, inserting a 0 on the left and dropping the last bit. |
| spread_first_bit | Replace every position with the first bit of the input. |
| spread_last_bit | Replace every position with the last bit of the input. |
| swap_halves | Swap the left and right halves of the string. |
| swap_pairs | Swap each adjacent pair of bits (positions 0/1, 2/3, ...). |
| xor_with_s0 | Can only be applied after another primitive. Computes the logical XOR between the original input $s_0$ and the output of the first primitive. |

Table 1: The 30 unary primitives used to construct functions in $F$.

## A.2 Table of Functions

The following table describes the one hundred specific compositions of primitives used to construct the evaluation suite in §3.2.

| Function | Function | Function |
|---|---|---|
| identity | spread_last_bit | swap_halves → shift_left_zero |
| rotl1 | invert_prefix | swap_halves → shift_right_zero |
| reverse_bits | invert_suffix | shift_left_zero → swap_halves |
| flip_bits | meta_constant | shift_right_zero → swap_halves |
| swap_halves | flip_bits → reverse_bits | keep_even_positions → flip_bits |
| majority | rotl1 → reverse_bits | keep_odd_positions → flip_bits |
| minority | reverse_bits → rotl1 | flip_bits → keep_even_positions |
| parity_fill | rotl1 → flip_bits | flip_bits → keep_odd_positions |
| alternating_start_one | swap_halves → reverse_bits | edge_mask → flip_bits |
| alternating_start_zero | swap_halves → flip_bits | center_mask → flip_bits |
| left_half | double_rotl → flip_bits | shift_left_zero → keep_even_positions |
| right_half | rotr1 → flip_bits | shift_left_zero → keep_odd_positions |
| double_rotl | spread_first_bit → flip_bits | shift_right_zero → keep_even_positions |
| rotr1 | spread_last_bit → flip_bits | shift_right_zero → keep_odd_positions |
| double_rotr | left_half → flip_bits | keep_even_positions → reverse_bits |
| ones_if_palindrome | right_half → flip_bits | keep_odd_positions → reverse_bits |
| mirror_half | flip_bits → left_half | shift_left_zero → parity_fill |
| spread_first_bit | flip_bits → right_half | shift_right_zero → parity_fill |
| spread_last_bit | double_rotl → reverse_bits | parity_fill → shift_left_zero |
| invert_prefix | rotl1 → swap_halves | parity_fill → shift_right_zero |
| invert_suffix | xor_with_s0 | spread_first_bit → shift_left_zero |
| meta_constant | flip_bits → xor_with_s0 | spread_last_bit → shift_right_zero |
| shift_left_zero | ones_if_palindrome → flip_bits | spread_first_bit → keep_even_positions |
| shift_right_zero | flip_bits → mirror_half | spread_last_bit → keep_odd_positions |
| swap_pairs | invert_prefix → reverse_bits | spread_first_bit → edge_mask |
| keep_even_positions | left_half → reverse_bits | spread_last_bit → edge_mask |
| keep_odd_positions | right_half → reverse_bits | spread_first_bit → center_mask |
| edge_mask | parity_fill → flip_bits | spread_last_bit → center_mask |
| center_mask | rotl1 → spread_first_bit | rotl1 → shift_left_zero |
| xor_with_s0 | shift_left_zero → flip_bits | rotl1 → shift_right_zero |
| flip_bits → reverse_bits | shift_right_zero → flip_bits | shift_left_zero → rotl1 |
| rotl1 → reverse_bits | flip_bits → shift_left_zero | shift_right_zero → rotl1 |
| reverse_bits → rotl1 | flip_bits → shift_right_zero | reverse_bits → edge_mask |
| rotl1 → flip_bits | swap_pairs → flip_bits | reverse_bits → center_mask |
| swap_halves → reverse_bits | shift_left_zero → reverse_bits | edge_mask → shift_left_zero |
| swap_halves → flip_bits | shift_right_zero → reverse_bits | edge_mask → shift_right_zero |
| double_rotl → flip_bits | spread_first_bit → flip_bits | shift_left_zero → shift_left_zero |
| rotr1 → flip_bits | shift_left_zero → edge_mask | shift_left_zero → swap_pairs |

Table 2: The complete set of 100 functions in $F$, consisting of 30 single primitives and 70 composed functions $((f \rightarrow g)(x) = g(f(x)))$.

## A.3 BitLoad of Functions

| Function / Composition | BitLoad | Function / Composition | BitLoad |
|---|---|---|---|
| identity | 8 | rotl1 | 8 |
| reverse_bits | 8 | flip_bits | 8 |
| swap_halves | 8 | majority | 8 |
| minority | 8 | parity_fill | 8 |
| alternating_start_one | 8 | alternating_start_zero | 8 |
| left_half | 4 | right_half | 4 |
| double_rotl | 8 | rotr1 | 8 |
| double_rotr | 8 | ones_if_palindrome | 8 |
| mirror_half | 4 | spread_first_bit | 1 |
| spread_last_bit | 1 | invert_prefix | 8 |
| invert_suffix | 8 | meta_constant | 0 |
| flip_bits → reverse_bits | 8 | rotl1 → reverse_bits | 8 |
| reverse_bits → rotl1 | 8 | rotl1 → flip_bits | 8 |
| swap_halves → reverse_bits | 8 | swap_halves → flip_bits | 8 |
| double_rotl → flip_bits | 8 | rotr1 → flip_bits | 8 |
| spread_first_bit → flip_bits | 1 | spread_last_bit → flip_bits | 1 |
| left_half → flip_bits | 4 | right_half → flip_bits | 4 |
| flip_bits → left_half | 4 | flip_bits → right_half | 4 |
| double_rotl → reverse_bits | 8 | rotl1 → swap_halves | 8 |
| xor_with_s0 | 0 | flip_bits → xor_with_s0 | 0 |
| ones_if_palindrome → flip_bits | 8 | flip_bits → mirror_half | 4 |
| invert_prefix → reverse_bits | 8 | left_half → reverse_bits | 4 |
| right_half → reverse_bits | 4 | parity_fill → flip_bits | 8 |
| rotl1 → spread_first_bit | 1 | shift_left_zero | 7 |
| shift_right_zero | 7 | swap_pairs | 8 |
| keep_even_positions | 4 | keep_odd_positions | 4 |
| edge_mask | 2 | center_mask | 6 |
| shift_left_zero → flip_bits | 7 | shift_right_zero → flip_bits | 7 |
| flip_bits → shift_left_zero | 7 | flip_bits → shift_right_zero | 7 |
| swap_pairs → flip_bits | 8 | shift_left_zero → reverse_bits | 7 |
| shift_right_zero → reverse_bits | 7 | swap_halves → shift_left_zero | 7 |
| swap_halves → shift_right_zero | 7 | shift_left_zero → swap_halves | 7 |
| shift_right_zero → swap_halves | 7 | keep_even_positions → flip_bits | 4 |
| keep_odd_positions → flip_bits | 4 | flip_bits → keep_even_positions | 4 |
| flip_bits → keep_odd_positions | 4 | edge_mask → flip_bits | 2 |
| center_mask → flip_bits | 6 | shift_left_zero → keep_even_positions | 4 |
| shift_left_zero → keep_odd_positions | 3 | shift_right_zero → keep_even_positions | 3 |
| shift_right_zero → keep_odd_positions | 4 | keep_even_positions → reverse_bits | 4 |
| keep_odd_positions → reverse_bits | 4 | shift_left_zero → parity_fill | 7 |
| shift_right_zero → parity_fill | 7 | parity_fill → shift_left_zero | 8 |
| parity_fill → shift_right_zero | 8 | spread_first_bit → shift_left_zero | 1 |
| spread_last_bit → shift_right_zero | 1 | spread_first_bit → keep_even_positions | 1 |
| spread_last_bit → keep_odd_positions | 1 | spread_first_bit → edge_mask | 1 |
| spread_last_bit → edge_mask | 1 | spread_first_bit → center_mask | 1 |
| spread_last_bit → center_mask | 1 | rotl1 → shift_left_zero | 7 |
| rotl1 → shift_right_zero | 7 | shift_left_zero → rotl1 | 7 |
| shift_right_zero → rotl1 | 7 | reverse_bits → edge_mask | 2 |
| reverse_bits → center_mask | 6 | edge_mask → shift_left_zero | 1 |
| edge_mask → shift_right_zero | 1 | shift_left_zero → shift_left_zero | 6 |
| shift_left_zero → swap_pairs | 7 | shift_left_zero → edge_mask | 1 |

Table 3: BitLoad for every primitive and composed function in $F$. Used in §4.3.

## B   A Global Metric Over All Programs

For a given function $f$, model $M$, and number of in-context examples $n$, we define the average accuracy over all context sets $E_N = \{E \subset S : |E| = n\}$:

$$A_f(M, n) = \frac{1}{|E_N|(|S| - n)} \sum_{E \in E_N} \sum_{x \in S \setminus E} A(f, E, x, M).$$

The overall benchmark score across all functions $F$ is then

$$P(M, n) = \frac{1}{|F|} \sum_{f \in F} A_f(M, n).$$

Because exact evaluation is intractable (for $n = 8$, $|E_N|(|S| - N) = \binom{|S|}{n}(|S| - n) = \binom{256}{8} \cdot 248 \approx 1.016 \times 10^{17}$) we estimate $A_f(M, n)$ via Monte Carlo, as discussed in §3.4.

## C   Full Accuracy Results

Here we show the full table of accuracies used in §4.1 to analyze the ICL capabilities of Evo2 and Qwen3 and perform the necessary statistical tests.

| Model | 1 Shot | 2 Shots | 4 Shots | 8 Shots | 16 Shots | 32 Shots | 64 Shots | 128 Shots |
|---|---|---|---|---|---|---|---|---|
| Qwen3 0.6B | $13.4_{\pm2.3}$ | $14.1_{\pm2.3}$ | $16.0_{\pm2.6}$ | $19.5_{\pm2.7}$ | $16.1_{\pm2.4}$ | $16.8_{\pm2.5}$ | $19.1_{\pm2.7}$ | $23.6_{\pm2.9}$ |
| Qwen3 1.7B | $13.1_{\pm2.2}$ | $15.0_{\pm2.7}$ | $\mathbf{17.2}_{\pm2.8}$ | $16.4_{\pm2.6}$ | $16.6_{\pm2.7}$ | $20.4_{\pm2.9}$ | $24.5_{\pm3.4}$ | $28.5_{\pm3.5}$ |
| Qwen3 4B | $13.5_{\pm2.4}$ | $15.2_{\pm2.4}$ | $17.0_{\pm2.6}$ | $15.4_{\pm2.5}$ | $18.4_{\pm2.7}$ | $24.8_{\pm3.2}$ | $24.5_{\pm3.4}$ | $29.4_{\pm3.4}$ |
| Qwen3 8B | $13.5_{\pm2.3}$ | $\mathbf{15.4}_{\pm2.5}$ | $16.9_{\pm2.7}$ | $19.6_{\pm2.7}$ | $20.6_{\pm2.8}$ | $\mathbf{27.9}_{\pm3.3}$ | $30.9_{\pm3.5}$ | $32.2_{\pm3.5}$ |
| Qwen3 14B | $\mathbf{14.0}_{\pm2.4}$ | $14.1_{\pm2.4}$ | $16.8_{\pm2.7}$ | $\mathbf{20.6}_{\pm2.9}$ | $\mathbf{22.9}_{\pm3.1}$ | $25.5_{\pm3.2}$ | $\mathbf{32.6}_{\pm3.7}$ | $\mathbf{33.8}_{\pm3.5}$ |
| Evo2 1B | $13.1_{\pm2.2}$ | $15.0_{\pm2.4}$ | $17.1_{\pm2.9}$ | $18.2_{\pm2.7}$ | $22.0_{\pm2.9}$ | $23.8_{\pm2.8}$ | $26.5_{\pm3.0}$ | $30.8_{\pm3.1}$ |
| Evo2 7B | $14.5_{\pm2.4}$ | $\mathbf{15.8}_{\pm2.7}$ | $17.6_{\pm2.9}$ | $\mathbf{22.6}_{\pm2.9}$ | $25.2_{\pm3.0}$ | $\mathbf{29.8}_{\pm3.1}$ | $32.2_{\pm3.1}$ | $41.0_{\pm3.4}$ |
| Evo2 40B | $\mathbf{15.0}_{\pm2.6}$ | $15.5_{\pm2.5}$ | $\mathbf{17.6}_{\pm2.8}$ | $21.6_{\pm3.1}$ | $\mathbf{26.4}_{\pm3.1}$ | $28.9_{\pm3.1}$ | $\mathbf{34.5}_{\pm3.2}$ | $\mathbf{41.1}_{\pm3.3}$ |

Table 4: In-context learning performance across model families and shot counts. Values show accuracy $\pm$ standard error. All numbers are percentages. **Bold** numbers show the best performance within a model family. Models are ordered by parameter count within each family.

## D   Meta-Regression for ICL Efficacy with Number of Demonstrations

We perform linear regressions to predict accuracy from shot count with each model. For each model $M$, fit the linear regression: $\hat{P}(M, n) = \alpha_0(M) + \alpha_1(M) \log(n) + \varepsilon$. The raw regressions are shown in Fig. 5a. Predictably, all $\alpha_1$ are positive as all models are capable of learning in-context.

We can interpret $\alpha_0$ as representing the model's base accuracy at the task, what it would logically achieve with only one shot to identify the task. We can then interpret $\alpha_1$ as the model's ICL efficacy: the speed at which it adapts to the task being presented and at which its accuracy improves. Analyzing how these values change across parameter values reveals insights into the ICL abilities of both the Qwen3 and Evo2 models.

First, we analyze how $\alpha_0$ changes in each model family – this analysis can be seen in Fig. 5b. Both Evo2's and Qwen3's initial $\alpha_0$ remains essentially constant model-to-model, indicating that all models have similar levels of few-shot baseline performance. Notably, Evo2 and Qwen3 have essentially identical intercepts at around 0.12. This implies that despite drastically different training data, the overall amount of prior knowledge the models have coming into this task is roughly similar. This rules out Qwen simply having 'less experience' with this sort of task.

If one looks at $\alpha_1$ – ICL efficacy – in Fig. 5c, a dramatically different picture is painted. Here, both Qwen3 and Evo2 follow similar patterns with a significant difference. Qwen3's ICL efficacy increases monotonically

with respect to parameters, more than doubling from the 0.6B to the 14B. Evo2 follows suit (albeit less dramatically), with ICL efficacy monotonically increasing from the 1B to the 40B.

In absolute terms, however, Evo2, when parameter-matched, adapts in-context faster than Qwen3 does. Evo2 40B outperforms Qwen3 14B significantly, and it takes until Qwen3 8B to exceed the ICL ability of Evo2 1B.

Taken together, this data suggests that Qwen3 and Evo2 have similar amounts of pretraining exposure to be able to solve these tasks, and that Evo2 simply has better overall ICL capability (in this regime) – even though Qwen's ICL ability increases more rapidly with respect to parameters.

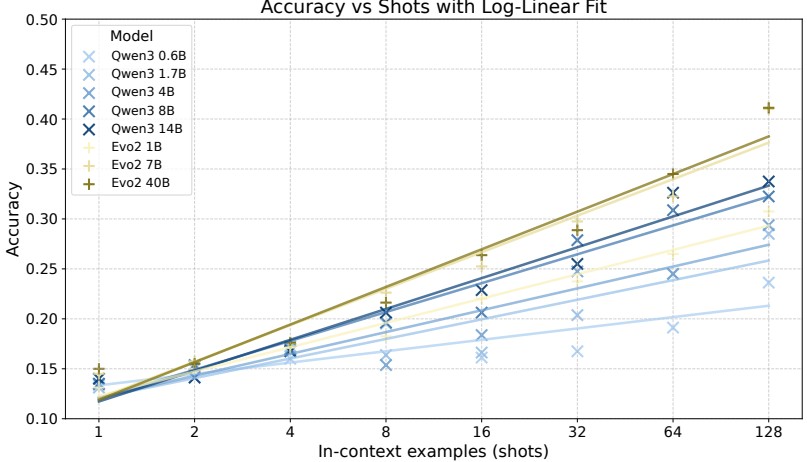

(a) Few-shot performance of Qwen3 and Evo2 models. All models show consistent linear improvement with respect to log(shots). In contrast, no such improvement occurs for the naive baseline.

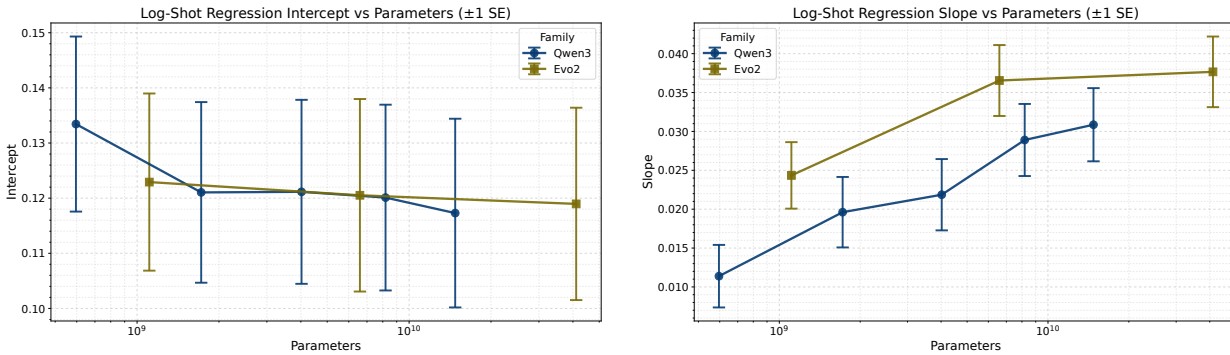

(b) Baseline accuracy decreases slightly with scale as the model gains more parameters for both Evo2 and Qwen3.

(c) ICL rate vs. model size: sharp gains up to 4B for Qwen3; mild boost from Evo2 1B to 7B; both plateau after 4–7B.

Figure 5: Few-shot behavior and scaling trends across Qwen3 and Evo2.

# E   Demonstration of Evo2's ICL Ability on a Genomic Task

We demonstrate that Evo2 7B can perform few-shot in-context learning (ICL) on human_nontata_promoters, a binary genomic classification task in which the model must predict whether a 251 nucleotide-long human DNA sequence is a promoter. To make the task non-trivial, sequences in the positive class are restricted to non-TATA promoters, removing the "TATA box", a pattern that serves as an obvious, exploitable marker (Umarov & Solovyev, 2017; Grešová et al., 2023).

For each test query, we construct a few-shot prompt by concatenating balanced examples from the training set (equal numbers of promoters and non-promoters), followed by a held-out test sequence. Each training example is immediately followed by an encoded label. The label consists of a 24 nucleotide buffer (a single nucleotide repeated 24 times) and then a 24 nucleotide label run (another nucleotide repeated 24 times). The buffer nucleotide and the two label nucleotides (for class 0 vs. class 1) are re-sampled uniformly at random without replacement from $\{A, C, G, T\}$ for every trial, so the mapping changes across evaluations. This prevents any nucleotide-specific biasing effects.

To predict the test label, we score two candidate prompts - one where the test sequence is followed by the class-0 label and one followed by the class-1 label - and choose the label with lower perplexity on the label run (i.e., computed only over the final 24 label nucleotides, excluding the buffer). This lets us measure whether Evo2 can infer the label mapping from the few-shot context and apply it to the query sequence.

We observe monotonic improvement with respect to shot count, with accuracy rising from below random baseline at 1-shot (48.2%) to 63.6% at 16-shot. These gains further rise to 68.6% after 256 shots. A saturated power law achieves an $R^2$ of 0.998 in the 2-256 shot regime, which is characteristic of few-shot ICL. Furthermore, we show via ablation that the improvement is due to correctly-labeled examples presented in context—randomly permuting the labels in the 16-shot regime drops performance from 63.6% back to random chance (50.2%).

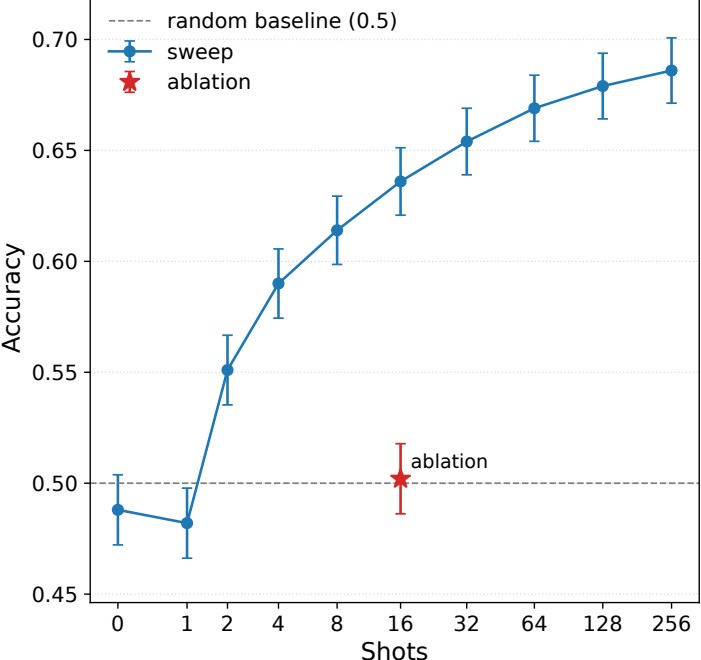

Figure 6: Performance of Evo2 7B on human_nontata_promoters at varying shot counts. Error bars are ± one standard error. Evo2 7B's performance shows clear monotonic improvements with respect to shot count.

# F    Analysis of ICL with Program Synthesis Tasks: Bit-Diversity

BitLoad captures a measure of theoretical information bottleneck required to make predictions in program synthesis tasks. However, this does not always correlate with what models find easy to do. In Fig. 8 and Fig. 9, we plot the accuracy of Evo and Qwen models with respect to BitLoad. We notice that few high BitLoad tasks have high accuracy compared to all medium BitLoad tasks, illustrating that some high BitLoad tasks can be easy for these models to solve, probably due to the nature of patterns found during pre-training.

To further analyze the nature of ICL exhibited through these tasks, we define BitDiversity as *the number of minority bits in the output string.* In Fig. 10 and Fig. 11, we plot the accuracy with respect to BitDiversity. These plots try to estimate the effect of entropy in the output on model performance, and we see a more expected trend: models tend to perform better on low-entropy outputs, regardless of BitLoad. However, bin-wise performance trends are always increasing with shots, supporting the central hypothesis that ICL increases with increasing number of demos in these models.

**Prevalence of $0$-BitDiversity outputs.**    Looking at the expected and predicted outputs of trials from our tasks (Fig. 12), we made a few interesting observations about 0-BitDiversity (BD) outputs.

- Around 25% of true targets are 0-BD. This is a significantly high number as we only have 2 0-BD bit strings in all possible bit strings of any length $K$. It implies that many of our tasks create low BitDiversity outputs on random inputs, i.e., 0-BD outputs.
- Models tend to produce a lot of 0-BD outputs, much higher than the number of 0-BD true targets in the low-shot regime. But this number quickly drops to the expected number with higher shots.
- A large portion of the baseline (1-shot) performance can be explained by this prevalence of 0-BD outputs, but with more shots we get stronger evidence of ICL with increasing correctly predicted non 0-BD cases.

**Understandable Mistakes**. A potential confound is what we will call "understandable mistakes". These mistakes occur when the model outputs an incorrect answer that would be correct for some other programs given the few-shot context. Formally, a model's output $y$ (see 3.4 for notation): $y = M\Big(e_1 \rightarrow f(e_1), e_2 \rightarrow f(e_2), \ldots, e_N \rightarrow f(e_N), x\Big)$ is an **understandable mistake** if $y \neq f(x)$ but there exists $f_2 \in F$ such that $\forall e_i \quad f(e_i) = f_2(e_i)$ and $y = f_2(x)$.

These understandable mistakes are an alarming confound at low shot counts, but their effect vanishes by $N = 16$. They occur most in tasks with low BitDiversity, which can often be confused with each other. Figure 7 below shows how understandable mistakes in Qwen3-4B's inferences decay exponentially as the number of shots increases.

**Noise due to Monte Carlo Trials**. We use $m = 8$ for our per-function Monte Carlo trials when we compute accuracy. This has little aggregate impact when comparing model performance across the entire suite, but drastically reduces the significance of results when comparing models and trends at the task-level. We only have eight discrete bands of accuracy at which we can estimate a model's per-task performance – which reduces the expressiveness of regression. Worse, this can lead to models getting a lucky prompt or two with a 0-BD output, which raises performance to 12.5% or 25% without the model truly understanding the task. Addressing these confounds would simply require increasing $m$ by an order of magnitude or so. Alternatively, tasks of interest could be identified and $m$ selectively increased for those tasks to enable more nuanced analysis.

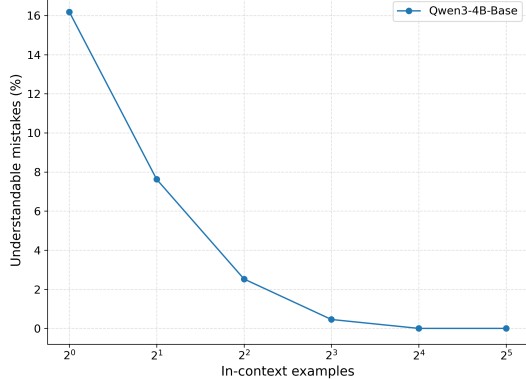

Figure 7: Qwen3-4B's rate of understandable mistakes with respect to the number of shots. Despite starting at 16% in the one-shot regime, they fall to less than 1% by 8 shots and vanish entirely at 32 shots. This underscores how understandable mistakes are only a confound at very low shot counts and can essentially be ignored past 4 shots.

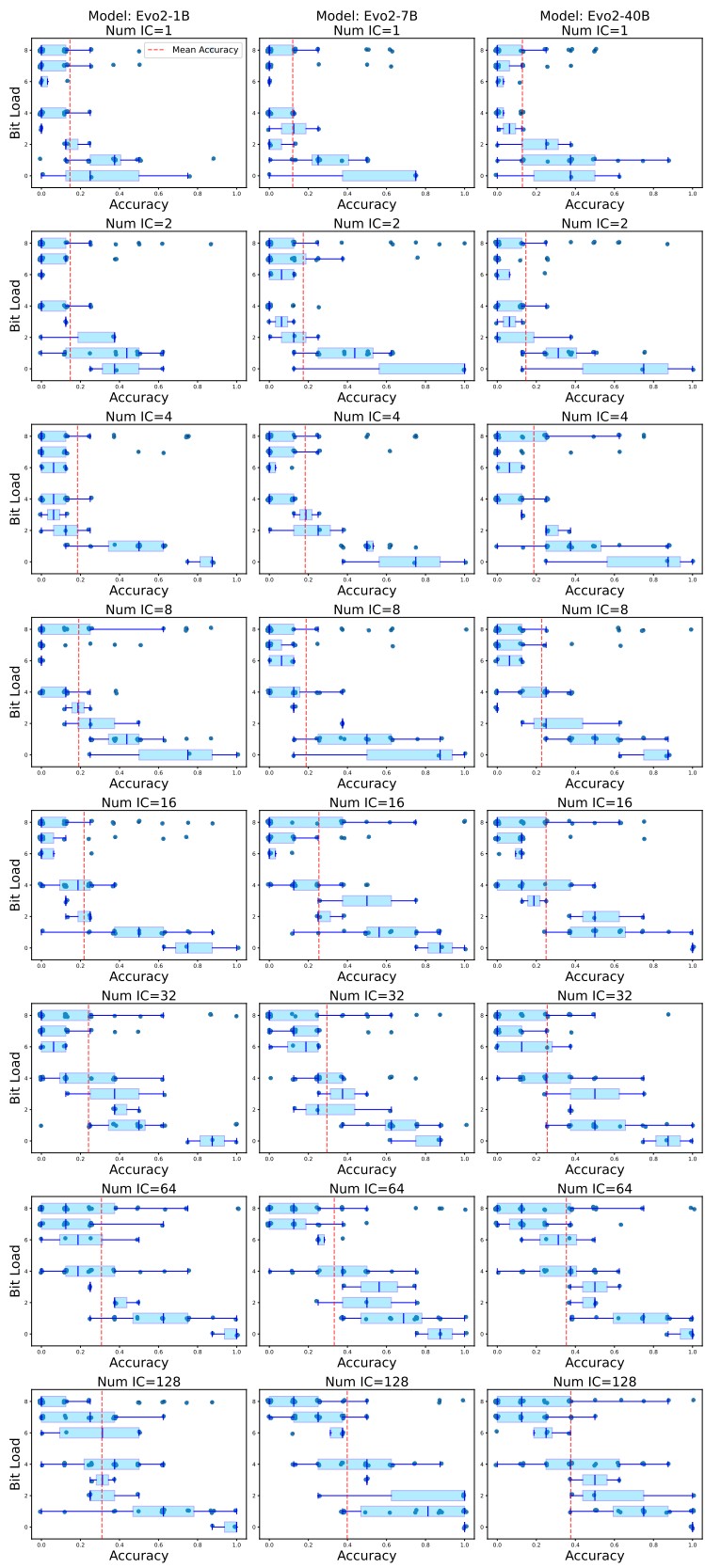

Figure 8: BitLoad vs Accuracy for all Evo models at all ICL shot-numbers. The per-BitLoad distribution of model performance at each shot level shows an uneven affinity of models for solving tasks at different BitLoads. However, all trends support our overall hypotheses.

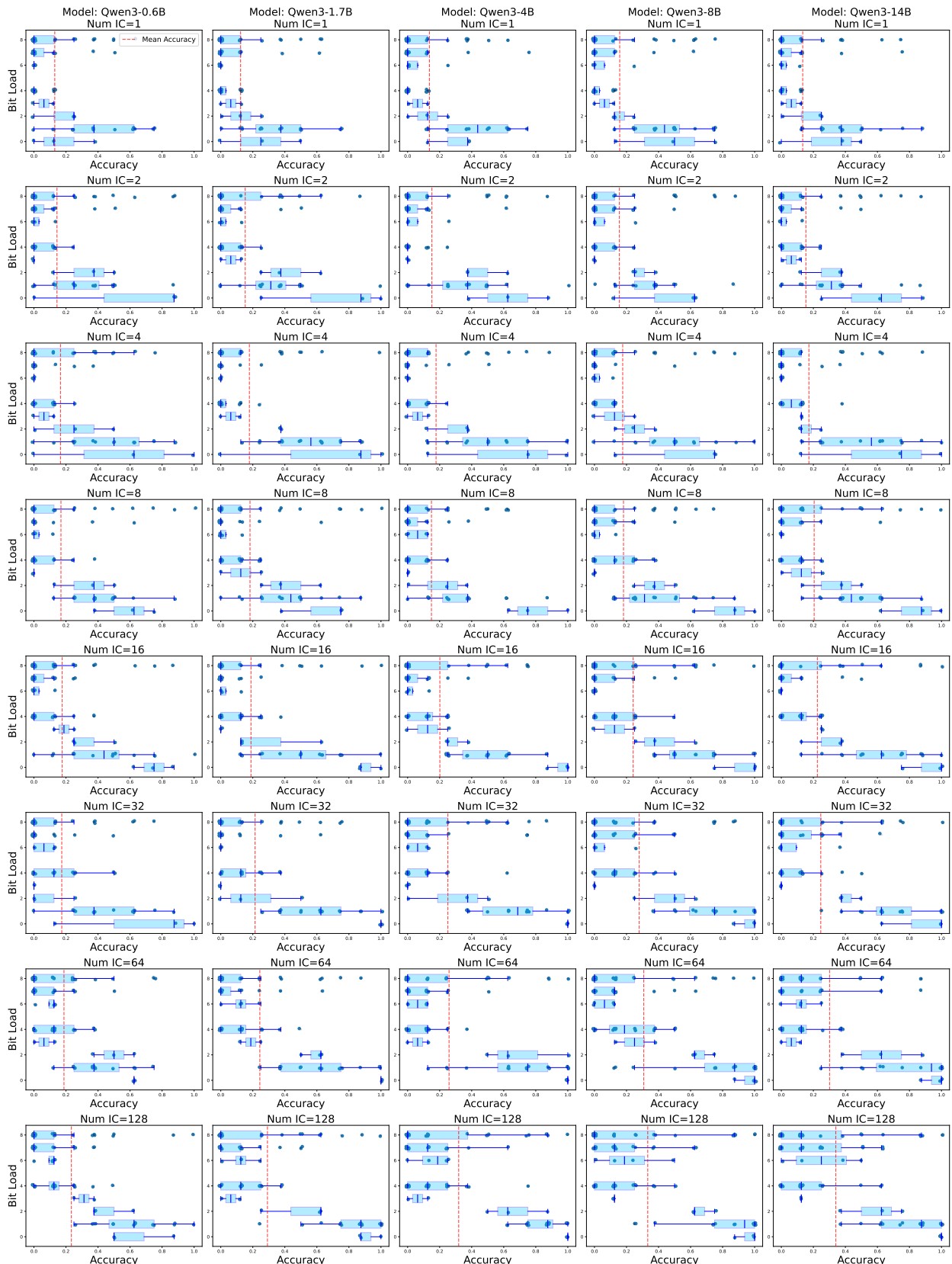

Figure 9: BitLoad vs Accuracy for all Qwen models at all ICL shot-numbers.

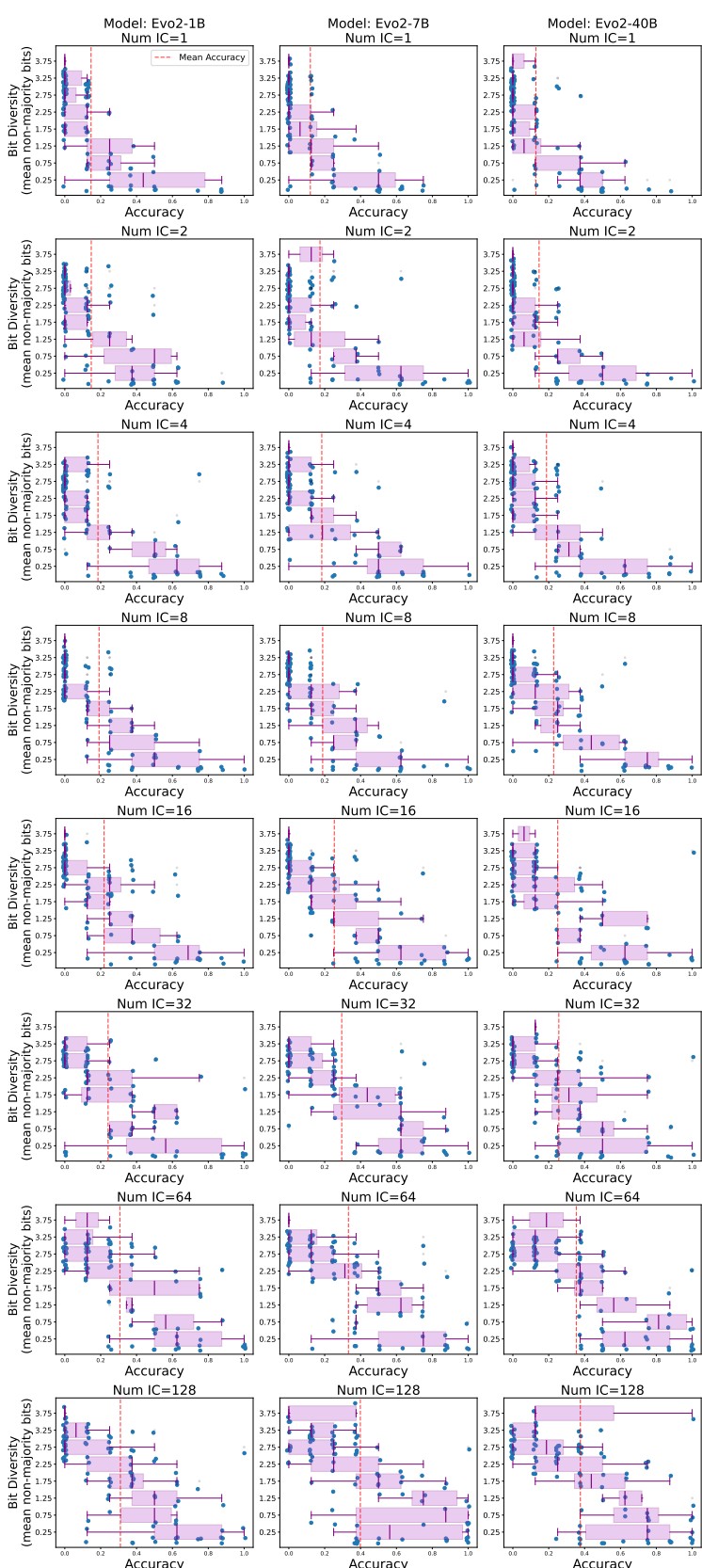

Figure 10: BitDiversity vs Accuracy for all Evo models at all ICL shot-numbers. Model performance follows a more natural pattern of increasing performance with decreasing output entropy. This highlights that it is difficult for models to decipher ICL patterns for high entropy outputs.

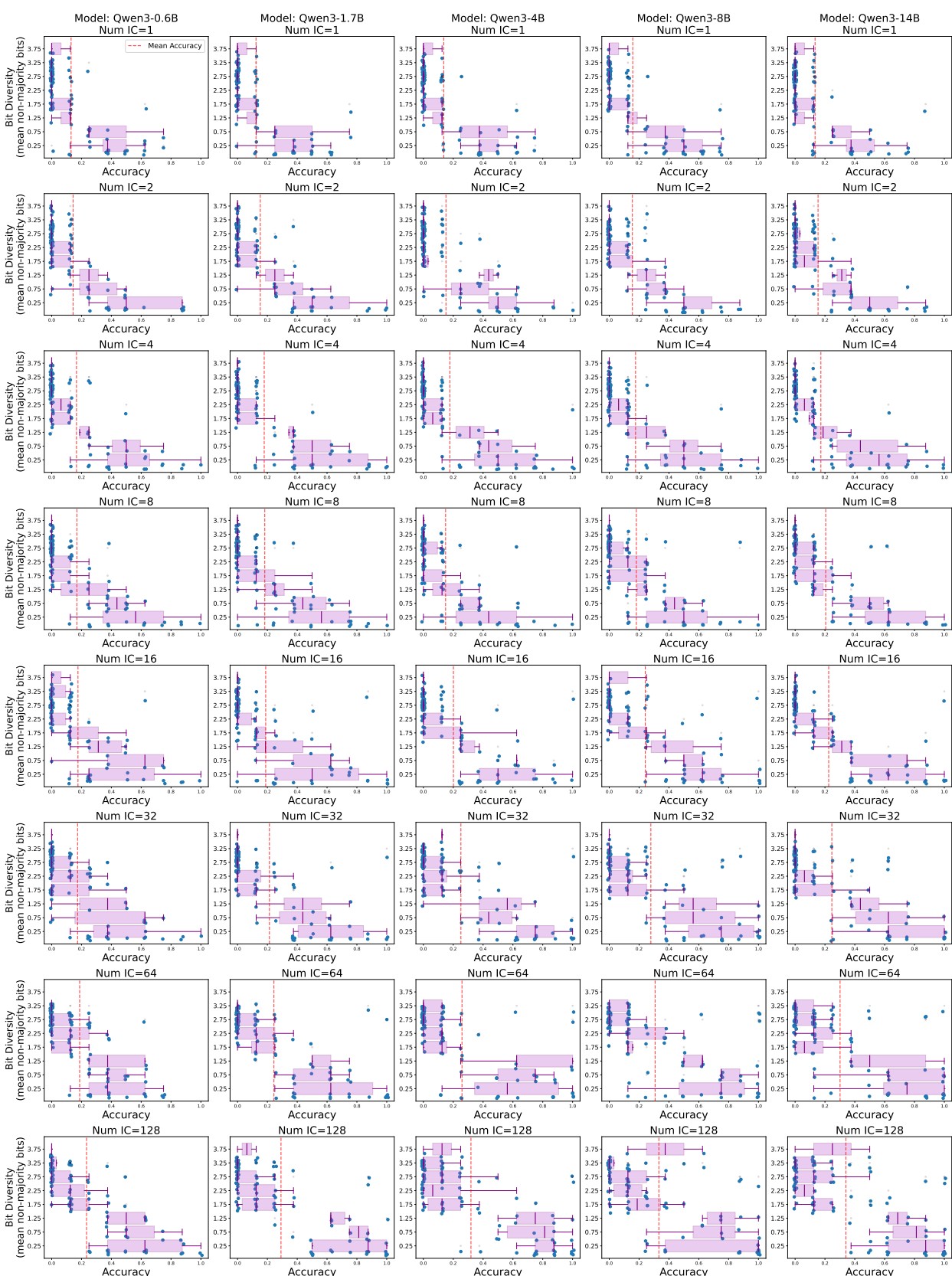

Figure 11: BitDiversity vs Accuracy for all Qwen models at all ICL shot-numbers.

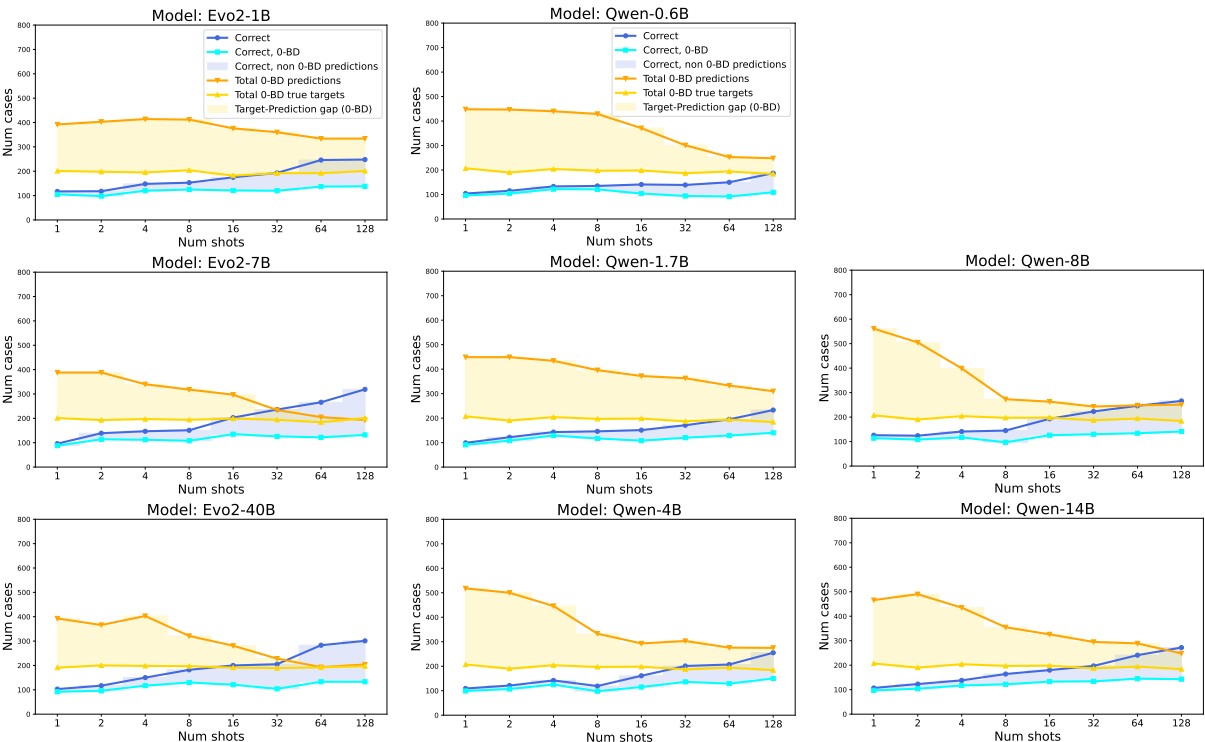

Figure 12: Enumerating cases of 0 BitDiversity. For our defined tasks, around 25% of the true targets have 0 BitDiversity (BD). With low-shots, models tend to produce a large number of 0-BD predictions. But this number decreases significantly with increasing shots and tends to match the actual prior value. Similarly, a majority of the baseline (1-shot) performance of models can be explained through 0-BD outputs. With a higher number of shots, the model starts learning higher entropy patterns and presents stronger evidence of ICL.

