# OpenReview forum: "Genomic Next-Token Predictors are In-Context Learners"
_TMLR — Accepted by TMLR_

### Review · Reviewer_yVdW · 2025-12-12

**Summary Of Contributions:**

Summary

This paper investigates whether emergent in-context learning (ICL) can arise in a genomic sequence modeling setting purely from large-scale next-token prediction. The authors study Evo2, a genomic foundation model trained primarily to predict the next nucleotide (A/C/G/T). To enable a controlled comparison with language models, they design a domain-agnostic symbolic pattern induction benchmark: the same underlying bitstring-to-bitstring transformations are rendered either as “language-like” digit strings or as “genome-like” nucleotide strings with randomized mappings.

Across model families and sizes, they find ICL-style scaling: accuracy improves approximately linearly with log (number of demonstrations). They further report qualitative differences in how performance scales with demonstrations vs model size, suggesting Evo2 benefits strongly from additional in-context examples under this encoding, while the language models benefit more from scaling model capacity.

Strengths

1. Novel cross-domain angle on ICL: The work provides a compelling existence proof that ICL-like behavior is not exclusive to human language training data. By evaluating next-token predictors in a non-linguistic symbolic modality (genomics), the paper helps disentangle whether ICL is a consequence of natural language semantics/compositionality or a more general property of sequence prediction at scale, and it connects naturally to “AI for science” motivations.

2. Careful, controlled experimental design: The benchmark construction is well-motivated: tasks are abstract, deterministic transformations, and the randomized digit/nucleotide mappings reduce trivial memorization. The evaluation includes a simple baseline and uncertainty estimates, making the reported trends easier to trust.

Weaknesses

1. Cross-model differences are not sufficiently contextualized: The paper would benefit from a clearer, more explicit comparison of the evaluated model families (architecture, tokenizer/vocabulary regime, context length, and training distribution). These factors likely affect performance on low-vocabulary symbolic strings and could partially explain observed differences between Evo2 and language baselines.

2. “Linguistic” modality may be an interface mismatch: Although digit strings are a practical rendering of bitstring tasks, it is debatable whether this constitutes a fair “linguistic” test. In particular, the language model’s performance may be limited by the prompt encoding/interface rather than a lack of ICL ability. Additional prompt/encoding variants would strengthen the claim that differences reflect underlying capability rather than representation alignment.

**Audience:**

Yes

**Audience Explanation:**

I think this paper both benefit people interested in AI for science, machine learning, and NLP.

**Claims And Evidence:**

Yes

**Claims Explanation:**

Yes, they introduce a controlled bitstring-to-bitstring symbolic induction benchmark that can be rendered in both “genomic-like” and “language-like” formats, and measure performance as the number of in-context demonstrations increases. Across settings, accuracy improves systematically with more demonstrations, and the reported trends provide clear empirical support for the paper’s central hypothesis.

**Requested Changes:**

As seen in weaknesses:

1. Clarify the difference between the two foundation models, genomic and linguistic. Discussing the impact of varying factors on the final in-context learning performance.

2. Explain the fairness of linguistic encoding by using digits in the experiment setting. Evo2 relies on a tiny alphabet (A/C/G/T) and is trained entirely on long contexts over that alphabet; Qwen3 is not. Even with digit-token parity, the training distribution matched to “weird symbolic strings” likely differs substantially.

---

> ### Author Response · Authors · 2026-01-18
>
> Thank you for the thorough review and the detailed feedback. We appreciate you highlighting the strength of our empirical results.  Regarding your concerns:
>
> > The paper would benefit from a clearer, more explicit comparison of the evaluated model families  (architecture, tokenizer/vocabulary regime, context length, and training distribution)
>
> **TLDR: Updated the paper with relevant information. Qwen3 and Evo2 have similar context lengths and have similar tokenization regimes in our experiment, but differ in architecture and training distribution.**
>
>  - Qwen3 is based on a conventional Llama-like transformer architecture, while Evo2 uses the StripedHyena2 architecture which intersperses convolutional layers with attention-based ones.
>
>  - Qwen3 uses a standard BPE tokenizer with a vocabulary size of 151,669. Evo2 uses a byte-level tokenizer, so individual nucleotides are mapped to individual tokens. Notable is that Qwen’s tokenizer maps single digits to single tokens.
>
>  - Qwen3’s 0.6B and 1.7B models have a context length of 32K. The remaining dense models have a context length of 128K. Evo2’s 1B model has a context length of 8K, and the remaining models have a context length of 1M.
>
>  - All Qwen3 models are trained on 36 trillion text tokens covering a wide variety of topics and languages. Evo2 7B is trained on 2.4 trillion tokens, and Evo2 40B is trained on 9.3 trillion tokens.
>
> We have updated the paper with this information (changes shown in red).
>
> ---
> > Clarify the difference between the two foundation models, genomic and linguistic. Discussing the impact of varying factors on the final in-context learning performance.
>
> **TLDR: Evo2’s architecture and training distribution may explain its superior performance to Qwen3 in this setting. Added this discussion to paper.**
>
> The downstream implications of the differences we mentioned above vary. The difference in context lengths is not likely to be relevant as our experiments do not approach the boundaries of any model’s context length. The tokenizer differences are unlikely to be relevant as Qwen3 tokenizes each digit as a singular token, akin to how Evo2 maps one nucleotide to a single token.
>
> The architecture differences may partially explain the difference in performance, as the Evo2 paper found the StripedHyena2 architecture to significantly outperform a vanilla transformer on long DNA sequences. This could give Evo2 an innate advantage on long contexts containing the same few symbols vs Qwen3.
>
> This advantage is further intensified by the training distribution. Though Evo2’s is trained on less tokens total, all of Evo2’s training tokens are long sequences of repeated nucleotides, and very little of Qwen3’s training tokens are long sequences of repeated digits. This gives Evo2 an inherent advantage.
>
> We added this discussion in the paper (changes shown in red).
>
> ---
> > Explain the fairness of linguistic encoding by using digits in the experiment setting. Evo2 relies on a tiny alphabet (A/C/G/T) and is trained entirely on long contexts over that alphabet; Qwen3 is not. Even with digit-token parity, the training distribution matched to “weird symbolic strings” likely differs substantially.
>
> **TLDR: We address this in the paper, in the last paragraph of section 4.4 - our setup is likely unfair to Qwen3. Our primary hypothesis was to study the existence of ICL in Evo2. Its relative performance to Qwen3 is subject to differences in pretraining.**
>
> As we agreed above, this is highly likely - Evo2’s pretraining better primes it for this sort of ICL. We address this directly in our paper, in the last paragraph of section 4.4 where we explicitly warn against using our results to imply Evo2’s ICL ability is superior to Qwen3’s, and concede that our few-shot prompting setup may be biased toward Evo2. The main trouble with making the bitstrings more in-distribution for Qwen (using 0s and 1s instead of random digits, for instance) is that *Qwen has pretrained knowledge of bit manipulation tasks and this would likely access them*. In fact, we  saw this in preliminary experiments - when presented in an intuitive format, with ones and zeroes, arrows separating inputs and outputs, and newlines - Qwen models performed far better than Evo models at these tasks. When we switched to the digit-based encoding, the performance gap between the models dropped to what it is now - mild, not drastic.
>
> ---
> **Thanks again for your careful review of our work. We hope our response has addressed your comments and look forward to addressing any remaining questions during the discussion period.**

---

> > ### Comment · Reviewer_yVdW · 2026-01-31
> >
> > Thanks for your response!

---

### Review · Reviewer_6Sjp · 2026-01-05

**Summary Of Contributions:**

In this paper, the authors are interested in the question: is "in-context learning" (ICL) unique to human language, or does it naturally emerge whenever a big model is trained to predict patterns? To get to the bottom of this, they ran a controlled experiment comparing a DNA model (Evo2) against a standard LLM (Qwen3), using abstract bitstring puzzles that could be translated into both genetic code and text. They found that the genomic model learned from in-context examples just like an LLM does, with performance improving reliably as it saw more demonstrations. This effectively suggests that the ability to learn on the fly isn't a special feature of human language, but rather a universal byproduct of training large models to compress complex, structured data. This is surprising, and might be theoretically significant, to see that a model trained solely on biological sequences picks up "in-context learning" behavior similar to that of a chatbot.

**Audience:**

Yes

**Audience Explanation:**

There is a huge ongoing debate about why in-context learning happens (is it the objective or the language?). By providing evidence that ICL emerges in a non-linguistic domain (DNA) purely through scale, this paper offers a crucial data point that separates the "magic of language" from the general mechanics of sequence modeling.

**Claims And Evidence:**

Yes

**Claims Explanation:**

The authors support their primary claim through a rigorous, controlled experimental setup. By abstracting "reasoning" into symbolic bitstring tasks (mapped to nucleotides for Evo2 and digits for Qwen3), they successfully isolate the mechanism of in-context learning from the semantics of human language. The results in Figure 2 are particularly convincing: they show a clear, log-linear improvement in accuracy as the number of demonstrations increases. The authors also attempt to match model parameter scales (e.g., comparing the 14B Qwen to the 40B Evo2). Section 4.3 (the BitLoad analysis) moves beyond simple accuracy metrics to show how the models fail (Qwen struggles with high-dependency tasks, Evo decays more slowly).

**Requested Changes:**

1. Demonstrate ICL on one actual genomic task using the same few-shot protocol. This would confirm that the ICL capability detected via bitstrings translates to the domain the model was actually built for.

---

> ### Author Response · Authors · 2026-01-18
>
> Thank you for the detailed review, and for highlighting the importance of our result! We address your concerns below:
>
> > Demonstrate ICL on one actual genomic task using the same few-shot protocol. This would confirm that the ICL capability detected via bitstrings translates to the domain the model was actually built for.
>
> **TLDR: We demonstrate that Evo2 can perform ICL on human_nontata_promoters, an actual genomic task**
>
> We evaluate Evo2 for ICL on [human_nontata_promoters](https://github.com/ML-Bioinfo-CEITEC/genomic_benchmarks), a binary genomic classification benchmark in which the model must predict whether a 251-nucleotide long human DNA sequence is a promoter. To make the task non-trivial, sequences in the positive class are restricted to non-TATA promoters, removing the “TATA box”, a pattern that serves as an obvious, exploitable marker.
> We use a similar few-shot prompting setup to our main paper, but switch to a label-based method for classification. We generate labels that consist of a 24 nucleotide buffer (a single nucleotide repeated 24 times) and then a 24 nucleotide label run (another nucleotide repeated 24 times). To predict the test label, we score two candidate prompts - one where the test sequence is followed by the class-0 label and one followed by the class-1 label - and choose the label with lower perplexity on the label run (i.e., computed only over the final 24 label nucleotides, excluding the buffer). This lets us measure whether Evo2 can infer the label mapping from the few-shot context and apply it to the query sequence.
> | Shots | Accuracy (± Stderr) |
> | --- | --- |
> | 0 | 0.4880 ± 0.0158 |
> | 1 | 0.4820 ± 0.0158 |
> | 2 | 0.5510 ± 0.0157 |
> | 4 | 0.5900 ± 0.0156 |
> | 8 | 0.6140 ± 0.0154 |
> | 16 | 0.6360 ± 0.0152 |
> | 32 | 0.6540 ± 0.0150 |
> | 64 | 0.6690 ± 0.0149 |
> | 128 | 0.6790 ± 0.0148 |
> | 256 | 0.6860 ± 0.0147 |
>
> We observe monotonic improvement with respect to shot count, with accuracy rising from below random baseline (50%) at 1-shot (48.2%) to 63.6% at 16-shot. These gains further rise to 68.6% after 256 shots.
>
> Further details are incorporated into the new draft we submitted (changes shown in red).
>
> ---
>
> **Thanks again for your careful review of our work. We hope our response has addressed your comments and look forward to addressing any remaining questions during the discussion period.**

---

### Review · Reviewer_xCj6 · 2026-01-16

**Summary Of Contributions:**

The authors compare in context learning (ICL) of large language models (LLMs) trained on natural language (Qwen) and genomic data (Evo2).

They design a common set of Boolean functions, and investigate how good Qwen and Evo2 are at learning them in context.

In both cases, ICL “works” in the sense that the larger the size of the context (or the size of the model), the better the accuracy. Interestingly, the improvements appear to occur at the same rate, log-linear in the number of in-context examples.

They also investigate which Boolean functions are easier to learn for Qwen and Evo2.

**Additional Comments:**

Isn't the "bitload" just the number of relevant features, often used in statistics and learning theory, e.g. in Nevo & Yaniv (2002)?

# Additional reference

Nevo & Yaniv, On Online Learning of Decision Lists, JMLR 2002

**Audience:**

Yes

**Audience Explanation:**

I think that the careful experimental design is interesting and original enough to have some appeal for the TMLR audience. I liked in particular the design of the Boolean functions, and the detailed analysis of the results.

One limitation of the paper is that there is not much insight to be gained about the quantitative comparison between language and DNA, because the two tokenizers are fundamentally different. This is acknowledged by the authors when they say

> Thus these results should be
taken as an existence proof of ICL in Evo2, not a definitive statement of Evo2 having more ICL ability than Qwen. We leave a comparison of these models across broader tasks to future work.

**Broader Impact Concerns:**

I have no concerns about the Broader Impact Statement written by the authors.

**Claims And Evidence:**

Yes

**Claims Explanation:**

All in all, the main claim (that ICL works at the same rate for both Qwen and Evo2) is well-supported by the experiments, with numerous statistical tests, in particular for the log-linear relationship. Confidence intervals on Fig 1.(4)  and Fig. 2 would be beneficial though.

I do have an issue with the sentence “To our knowledge, the only claimed instance of emergent ICL outside language is in vision: Bai et al. (2024)” (just before Section 3). I am no ICL expert but I believe there are several other prior examples :

-	On time-series: the chronos models (Ansari et al., 2024, 2025) and TimesFM (Das et al., 2025)
-	Bio-xLSTM, on biological sequences (Schmidinger et al., 2025)

The tasks involved there are quite different from the one considered in this submission, so this is not a big issue, but these papers should be discussed (in particular Bio-xLSTM that does experiments on DNA data).

# Additional references

- Ansari et al., Chronos: Learning the Language of Time Series, TMLR 2024
- Ansari et al., Chronos-2: From Univariate to Universal Forecasting, arXiv preprint arXiv:2510.15821, 2025
- Das et al., A decoder-only foundation model for time-series forecasting,  ICML 2024
- Schmidinger et al., Bio-xLSTM: Generative modeling, representation and in-context learning of biological and chemical sequences, ICLR 2025

**Requested Changes:**

- Add confidence intervals and discuss them (see above)
- Discuss prior work on ICL on non-text data

---

> ### Author Response · Authors · 2026-01-18
>
> Thank you for the thorough review and suggestion of so much relevant literature! We appreciate the care you put into your critique.  Regarding your concerns:
>
> > All in all, the main claim (that ICL works at the same rate for both Qwen and Evo2) is well-supported by the experiments, with numerous statistical tests, in particular for the log-linear relationship. Confidence intervals on Fig 1.(4) and Fig. 2 would be beneficial though.
>
> A table of bootstrapped standard errors is presently in Appendix C, Table 4. We didn’t add these standard errors to the figures as it increased clutter and made them harder to read. Let us know if this addresses your comment!
>
> ---
> > I do have an issue with the sentence “To our knowledge, the only claimed instance of emergent ICL outside language is in vision: Bai et al. (2024)” (just before Section 3). I am no ICL expert but I believe there are several other prior examples :
> On time-series: the chronos models (Ansari et al., 2024, 2025) and TimesFM (Das et al., 2025)
> Bio-xLSTM, on biological sequences (Schmidinger et al., 2025)
> The tasks involved there are quite different from the one considered in this submission, so this is not a big issue, but these papers should be discussed (in particular Bio-xLSTM that does experiments on DNA data).
>
> **TLDR—Chronos and TimesFM papers do *not* directly mention or investigate in-context learning. Chronos 2 investigates in-context learning, but does *not* perform few-shot scaling evaluations and relies extensively on synthetic data, which pushes it toward the ‘Meta-ICL’ regime. Bio-xLSTM is similar as it explicitly trains for in-context learning, which makes it meta-ICL as well. Our work is distinguished by showing emergent ICL.**
>
> Thank you for pointing these additional papers out!
>
> The Chronos and TimesFM papers study the zero-shot inference capabilities of next-token predictors, i.e., given a context of a given problem their models can predict the forthcoming values (tokens). This differs from the type of ICL we evaluate, which explicitly relies on few-shot learning of a transformation, not zero-shot adaptation. In fact, in Fig 2, we report that Evo2’s performance begins far below a naive mode baseline (simply guessing the most common output) at one shot, indicating very poor zero-shot capability. Evo2’s ability only becomes meaningful when many shots are provided. This makes the type of ICL we demonstrate qualitatively different from zero-shot ability, as it explicitly only appears in few-shot contexts.
>
> Chronos 2 does investigate the few-shot regime through cross-batch learning, where the model’s performance can be improved by learning from multiple time series provided in-context. However, their work is distinct from ours for multiple reasons - first, they rely exclusively on synthetic data to teach the model how to perform cross-batch learning: explicitly meta-ICL, not emergent ICL. Second, they only test the model zero-shot and few-shot - they don’t examine how performance changes across different shot counts. This significantly reduces the ability to deduce how the model is performing in-context learning, or if its in-context learning is similar to that exhibited by LLMs.
>
> Bio-xLSTM does investigate in-context learning, but they **explicitly train** models to perform it by creating pretraining data to teach few-shot inference and generation. This falls under our established framework of ‘Meta-ICL’ in our paper (discussed in related work) - training a model for ICL explicitly, not showing its emergence after pretraining done without ICL in mind. However, this is still a very relevant piece of literature and we have updated the paper to add a reference to it in the introduction and discuss what distinguishes our work.
>
> Thanks again for pointing out these references. We’re including the appropriate citations (changes shown in red), and necessary analysis to contextualize what makes our work distinct.
> (...cont in follow-up comment)

---

> > ### Author Response · Authors · 2026-01-18
> >
> > (...cont above)
> >
> > ---
> > > One limitation of the paper is that there is not much insight to be gained about the quantitative comparison between language and DNA, because the two tokenizers are fundamentally different. This is acknowledged by the authors when they say: “Thus these results should be taken as an existence proof of ICL in Evo2, not a definitive statement of Evo2 having more ICL ability than Qwen. We leave a comparison of these models across broader tasks to future work.”
> >
> > We want to clarify something here that we didn’t adequately explain in the original paper - the Qwen tokenizer maps each digit to *one* token, and the Evo2 tokenizer maps each nucleotide to *one* token. This allows us to directly compare ICL capability without tokenization as a confounding factor, as each atomic unit of the prompt (digit/nucleotide) is mapped to *one* token. We suspect differences in performance may be explained by training distribution and architecture, which we have acknowledged and have further updated the paper to discuss these differences in more detail. We ultimately think this doesn’t significantly detract from the paper’s significance as we set out to prove that Evo2 can perform ICL, not relate it directly to Qwen. We mainly compare to Qwen as a control, to ensure the dynamics of ICL are similar.
> >
> > ---
> > > Isn't the "bitload" just the number of relevant features, often used in statistics and learning theory, e.g. in Nevo & Yaniv (2002)?
> >
> > They are very similar, yes, though bitload is a bitstring-specific metric and ‘relevant features’ is more general. We have added a citation to this paper and reference that BitLoad is similar to this commonly used concept.
> >
> > ---
> >
> > **Thanks again for your careful review of our work. We hope our response has addressed your comments and look forward to addressing any remaining questions during the discussion period.**

---

> > > ### Comment · Reviewer_xCj6 · 2026-01-31
> > >
> > > Thanks for your clarifications!
> > >
> > > > A table of bootstrapped standard errors is presently in Appendix C, Table 4. We didn’t add these standard errors to the figures as it increased clutter and made them harder to read. Let us know if this addresses your comment!
> > >
> > > I would suggest at least adding a reference to the Appendix in the caption of Figure 2, and adding the confidence intervals to Fig. 1.4 (there is only one curve per plot, I don't think it will make it less readable).
> > >
> > > > Chronos and TimesFM papers do not directly mention or investigate in-context learning. Chronos 2 investigates in-context learning, but does not perform few-shot scaling evaluations and relies extensively on synthetic data, which pushes it toward the ‘Meta-ICL’ regime. Bio-xLSTM is similar as it explicitly trains for in-context learning, which makes it meta-ICL as well. Our work is distinguished by showing emergent ICL.
> > >
> > > Thanks! As I mentioned in my review, I'm not an expert of ICL, and after reading your answer and these papers, I agree. Thanks for updating the paper.
> > >
> > > > We want to clarify something here that we didn’t adequately explain in the original paper - the Qwen tokenizer maps each digit to one token, and the Evo2 tokenizer maps each nucleotide to one token.
> > >
> > > Thanks! Got it.

---

> > > > ### Author Response · Authors · 2026-01-31
> > > >
> > > > We added a reference to the Appendix in Fig 2, and added confidence intervals (95%) to Fig 1.4. Thank you for the feedback!

---

> > > > > ### Comment · Reviewer_xCj6 · 2026-02-03
> > > > >
> > > > > Thanks! It looks good.

---

### Decision · Action_Editor_Fk16 · 2026-02-21

**Recommendation:** Accept with minor revision

**Additional Comments:**

This is a strong paper that the reviewers also were positive about. The reviewers have suggested some very helpful improvements that have already made their way into the manuscript. I am very happy to accept it with a few minor revisions. I would almost support accepting as is, but I have a few cosmetic points and some literature pointers to bring up for the camera ready version.

- In sec. 3.1, Lewis & Mitchell 2024 is cited with the comment that they "rely on a richer vocabulary (e.g., shapes, colors, or linguistic tokens with explicit semantic roles)" ; I haven't checked the other papers in the citation but Lewis & Mitchell does not seem to fit this as they explicitly study purely symbolic low-vocabulary sequence tasks e.g. "Extend sequence: a b c d -> a b c d e". I was in fact preparing to recommend you take a look at this work and related work as additional context for your bit string encoding and for context for readers for additional biases of how llms or humans handle such tasks. Another perhaps relevant reference in this direction is https://arxiv.org/abs/2411.02348. Perhaps you could point to this line of work for completeness and reader navigation.

 - Please double-check your references to cite officially published versions instead of preprints when possible; e.g., Lewis & Mitchell 2024 was published in TMLR, there may be other instances.

Cosmetics:
 - Please use vector formats (pdf or svg) for graphs whenever possible, in particular such that labels be searchable and selectable and file sizes be smaller.
 - Please increase font sizes in Figure 2
 - The caption of Figure 4 does not have sufficient space and overlaps with paragraph text, please correct this.

**Audience:**

Yes

**Audience Explanation:**

The topic is undoubtedly relevant and timely; I am sure this paper will be found interesting by a substantial slice of the audience.

**Claims And Evidence:**

Yes

**Claims Explanation:**

It is my opinion that the paper sets up very clear and verifiable claims which are then adequately supported. The reviewers all agree and I would highlight that the strength of the evidence has improved during the review process through changes such as error bars and an additional experiment on a genomic task.

---

> ### Author Response · Authors · 2026-02-26
>
> Thank you for accepting our paper with minor revisions. We deeply appreciate your understanding of our work, and appreciate the time you have spent analyzing it. We address your concerns below - all mentioned changes have been applied to our submitted revision:
>
> > In sec. 3.1, Lewis & Mitchell 2024 is cited with the comment that they "rely on a richer vocabulary (e.g., shapes, colors, or linguistic tokens with explicit semantic roles)" ; I haven't checked the other papers in the citation but Lewis & Mitchell does not seem to fit this as they explicitly study purely symbolic low-vocabulary sequence tasks e.g. "Extend sequence: a b c d -> a b c d e". I was in fact preparing to recommend you take a look at this work and related work as additional context for your bit string encoding and for context for readers for additional biases of how llms or humans handle such tasks. Another perhaps relevant reference in this direction is https://arxiv.org/abs/2411.02348. Perhaps you could point to this line of work for completeness and reader navigation.
>
> We appreciate the concerns you raise here. However, Lewis & Mitchell’s tasks still rely on a vocabulary (either 26 letters, 9 numbers, etc.) far exceeding what one can encode with **four** nucleotides. Furthermore, they require intrinsic knowledge about letter/numbering ordering that a genomic model would not have. Furthermore, when evaluating out-of-distribution tasks, Lewis & Mitchell rely on linguistic instructions to convey the ordering of novel tokens. This has no meaningful analogue in genomic models. https://arxiv.org/abs/2411.02348 has similar limitations. We update this section with an additional clarification explaining why these analogical tasks are still not applicable to our domain, and clarify wording to better represent the work of Lewis & Mitchell.
>
> > Please double-check your references to cite officially published versions instead of preprints when possible; e.g., Lewis & Mitchell 2024 was published in TMLR, there may be other instances.
>
> We have double-checked our references and updated them where appropriate to reflect publications also featured in journals or at conference proceedings.
>
> > Please use vector formats (pdf or svg) for graphs whenever possible, in particular such that labels be searchable and selectable and file sizes be smaller.
> Please increase font sizes in Figure 2
> The caption of Figure 4 does not have sufficient space and overlaps with paragraph text, please correct this.
>
> We have implemented all these requested changes. Thank you for pointing them out.
>
> ---
>
> Again, thank you for accepting our paper and for these insightful comments to polish the final result.

---

> > ### Comment · Action_Editor_Fk16 · 2026-03-02
> > **Thank you**
> >
> > All looks good to me, thank you for the revision and the clarification.